



# The influence of burial history on physical properties of claystones – Overview of a systematic research program across scales

Raphael Burchartz[1*], Timo Seemann[1], Garri Gaus[2,4], Lisa Winhausen[1], Mohammadreza Jalali[1], Brian Mutuma Mbui[2], Sebastian Grohmann[2], Linda Burnaz[2], Marlise Colling Cassel[2], Jochen Erbacher[3], Ralf Littke[2] & Florian Amann[1,4]

[1]Chair of Engineering Geology and Hydrogeology, RWTH-Aachen University, Aachen, Germany
[2]Chair of Organic Biogeochemistry in Geo-Systems, RWTH-Aachen University, Aachen, Germany
[3] Federal Institute for Geosciences and Natural Resources, 30655 Hannover, Germany
[4] Fraunhofer Research Institution for Energy Infrastructures and Geotechnologies IEG, 52056 Aachen, Germany

*Correspondence to*: Raphael Burchartz (burchartz@lih.rwth-aachen.de)

## Abstract

The search for a suitable host rock for the deep geological disposal of high-level radioactive waste is one of the major geological challenges of our time. In Germany, alongside rock salt and crystalline rock, claystones are considered a promising geomaterial and the subject of intensive research within the scope of the site selection process. Particular focus is placed on those rock properties that are intended to prevent the migration of radionuclides into the environment effectively, referred to as barrier properties. These primarily include low permeability, self-sealing efficiency with respect to fractures, sorption capacity, and mechanical properties for long-term stability of the underground infrastructure.

However, these properties are dependent on numerous factors such as mineralogical composition, temperature and stress conditions and water content. Among these factors, the burial history and thus compaction affects porosity, permeability, and mechanical properties. Within the framework of the MATURITY project, the impact of burial history on barrier-relevant properties of claystones is investigated in a detailed multidisciplinary investigation approach across scales. For this purpose, a Lower Jurassic claystone formation (the Amaltheenton-Formation (Fm)) was investigated which was subjected to variable maximum depth and subsequent uplift during its burial history. Eight shallow boreholes at the margin area of the Lower Saxony Basin (Germany) were drilled through the formation with varying degrees of maturation. Comprehensive field and laboratory investigations are aimed to analyze burial-induced alterations of claystone barrier properties, and thereby advance the current understanding of these processes. With this contribution, we aim to establish a framework for a series of detailed parametric studies that will systematically approach the dependencies between burial history and petrophysical (e.g. density, permeability), geochemical (e.g. cation exchange capacity), hydrogeological (e.g. transitivity, storativity) and mechanical (e.g. rock strength, elasticity) properties.

We present the first results of different project steps that show (a) a relatively homogeneous clay dominated mineralogical composition of the Amaltheenton-Fm across the boreholes, (b) an increase of max. burial temperatures (83 °C-169 °C) over a lateral distance of ~50 km within the investigation area, (c) a gradual increase in bulk density accompanied by a reduction in



porosity and permeability for normally compacted Amaltheenton-Fm sequences along increasing max. burial temperatures, (d) a reverse trend of those parameters for a potentially under-compacted Amaltheenton-Fm sequence, and (e) hydraulic

conductivity determined from in-situ hydraulic tests that span two orders of magnitude ($10^{-5}$ m/s to $10^{-7}$ m/s).

## 1 Introduction

Claystones are considered as potential host rocks for the long-term disposal of high-level radioactive waste (HLW) in many countries. Famous examples of European HLW host rock candidates include the Opalinus Clay (OPA; Switzerland), the Boom Clay (Belgium), and the Callovo-Oxfordian argillite (COx, France) (Delage et al., 2010; Norris, 2017). Their suitability to act

as natural barriers is mainly due to their favorable physical matrix properties, such as very low permeability (down to $10^{-21}$ m$^2$), preventing significant focused fluid flow and related advective mass transport of radionuclides in aqueous solutions (OECD & Nuclear Energy Agency, 2022; Fisher et al., 2023). Additionally, their nuclide sorption capacity, and self-sealing behavior, mitigate the risk of radionuclide migration into the environment (Bastiaens et al., 2007; OECD & Nuclear Energy Agency, 2022). However, there are numerous factors influencing claystone properties and the related sealing integrity of potential host

rock formations. Physical and chemical properties of claystones are prone to burial-related alterations, mainly attributed to compaction due to changing in-situ stress and temperature conditions that might influence their barrier attributes (Fig. 1) (Aplin & Yang, 1995; Bjørlykke, 2006; Cripps & Czerewko, 2017; Ewy et al., 2020; Fisher et al., 2023). Further alteration of those properties might additionally be induced by processes such as uplift, erosion, and isostatic rebounding during the course of the burial history (Fink et al., 2019; Mazurek et al., 2023). Hence, changes in physical parameters related to burial history play a

crucial role for claystone assessment as natural barriers and host rocks for HLW disposal. The general impact of those changes and the mutual interdependencies must be thoroughly investigated to evaluate and predict the rock's sealing capacity over time scales of up to 1 Ma.

One of the fundamental changes in claystones associated with progressive burial is the gradual reduction of porosity and bulk volume, accompanied by an increase in density and elastic wave velocity (Athy, 1930; Aplin & Yang, 1995; Aplin et al., 2006;

Ewy et al., 2020; Fisher et al., 2023). The reduction of porosity in claystones is typically a result of the combined effects of mechanical and chemical compaction, such as the precipitation of carbonaceous or siliceous mineral phases (Jones & Addis, 1984; Addis & Jones, 1985; Jones & Addis, 1985; Broichhausen et al., 2005; Bjørlykke, 2006; Armitage et al., 2010; Ewy et al., 2020). In response to the increase of effective in-situ stress during burial, mechanical compaction controls dewatering processes, pore evolution (void reduction), and particle reorientation, serving as the dominant diagenetic mechanism at

relatively shallow depths (<2 km) and low temperatures (<70°C) (Aplin & Yang, 1995; Bjørlykke, 1998; Peltonen et al., 2009; Cripps & Czerewko, 2017). Further, porosity reduction is usually associated with chemical compaction such as mineral cementation and clay mineral reactions occurring during deeper burial and higher related temperatures (>70°C) (Bjørlykke, 1998). One of the most important diagenetic clay mineral reactions driven by temperature changes is the progressive





transformation of smectite into thermally more stable illite, occurring through an intermediate mixed-layer stage that forms
illite/smectite (I/S) mixed-layers (Pollastro, 1993; Berthonneau et al., 2017). This process is commonly referred to as illitization
and involves, besides porosity reduction, dehydration processes, reduced swelling potential, changes in the cation exchange
capacity (CEC) and grain sizes (Ohazuruike & Lee, 2023). However, porosity reduction during burial may be impeded by
overpressure generation as a result of compaction disequilibrium (under-compaction), hydrocarbon generation, or clay
diagenetic processes such as illitization, potentially leading to fracturing processes and fracture-related fluid flow, thus
impairing the rock's barrier function (Eaton, 1975; Bowers, 1995; Swarbrick & Osborne, 1998; Hart et al., 2023).

As clay muds are compacted to lower porosity claystones, permeability can decrease by six or more orders of magnitude (down
to $10^{-23}$ m²), improving their barrier function in HLW disposal (Neuzil, 2019). Quantification and characterization of claystone
permeability has long been recognized as a challenging, time-consuming task (Neuzil, 1994, 2019). A fundamental challenge
when quantifying permeability lies in the strong scale dependency (Neuzil, 1994, 2019; Van Der Kamp, 2001). Permeability
measurements on smaller-sized low permeable rocks tend to deliver a measure of the unfractured matrix permeability, while
secondary structures such as transmissive fractures remain neglected, even though fractures or other heterogeneities control
the larger-scale hydraulic behaviour (Neuzil, 1994; Van Der Kamp, 2001). Therefore, differences between the matrix and
effective permeability might differ by orders of magnitude (Van Der Kamp, 2001) and scale dependencies have to be
considered during repository planning.

Mechanical rock properties such as rock strength and elasticity play a major role in barrier integrity and stability considerations
for HLW host rocks. Relatively low rock strength of claystones and strength anisotropy due to their bedded structure may hold
considerable influence on the construction of underground facilities (Blümling et al., 2007; Wild & Amann, 2018a). Ductile,
soil-like claystones hold advantages in terms of self-sealing properties, but challenge the construction of a stable subsurface
structure at larger depth, while harder, more brittle claystones are vulnerable to fracture processes, significantly enhancing
preferential fluid pathways, e.g., in an excavation damage zone (Neuzil, 1994; Bossart et al., 2002, 2004). Various experimental
studies demonstrated that the mechanical behavior of claystones mainly depends on (a) the mineralogical composition, (b) the
porosity and water saturation, and (c) pressure and temperature conditions (Ibanez & Kronenberg, 1993; Aplin & Macquaker,
2011; Sone & Zoback, 2013; Rybacki et al., 2015; Amann et al., 2017; Busch et al., 2017; Cripps & Czerewko, 2017; Rutter
et al., 2017; Wild & Amann, 2018a, 2018b; Winhausen et al., 2022, 2023). At the formation/basin scale, the mechanical
behavior is therefore dependent on (a) the depositional environment and resulting mineralogical composition, and (b) the burial
history with related pressure and temperature conditions, systematically altering the physical and chemical properties from the
moment of deposition. In general, the mechanical behavior shows a transition from soil-like ductile to rock-like brittle behavior
with increasing burial depth due to various diagenetic mechanisms, such as i) closer grain-to-grain bonding of clay grains with
decreasing porosity and ii) the formation of cementing minerals, like carbonates or silicates (Ewy et al., 2020).





**Figure 1: Simplified and condensed alterations in claystones induced by changing in-situ stress and temperature conditions along their burial history: (a) the general burial history of claystones. Roman numbers represent essential processes along the burial history of claystones with I.=deposition, II.=mechanical compaction, III.=chemical compaction, and IV.=uplift (modified after: Lehocki and Avseth, 2020). The range bars and colored arrows mark ranges along the burial history in which the respective process contributes to alterations within the claystones. While mechanical compaction is only effective during burial, chemical compaction remains also active during uplift; (b) alterations in claystones induced by the processes I., II., III., and IV.. During deposition, clays show a random assemblage of crystallite aggregates. The porosity dependents on detrital clay composition and assemblage architecture. An increase in effective in-situ stress will lead to realignment and close packing of aggregates, reduced pore space, expulsion of pore water. Chemical compaction becomes the dominant compaction process when temperatures reach >70°C. Dissolution, cementation, and mineral reactions such as illite-smectite conversion will lead to further loss of porosity. Uplift will lead to decompaction, erosion, and fracturing (mainly along bedding); (c) and (d) show burial related trends in different claystone properties ((c) modified after Bjørlykke, 1998).**



It is essential to assess the barrier properties of claystone formations by investigating the relevant rock properties, along with their complex (inter-)dependencies as these are inherently linked to each other. While the general changes induced by the burial and uplift of claystone have been thoroughly investigated in recent decades, systematic and quantitative studies on the depth- and temperature-dependent progression of these changes remain scarce. The same holds true for studies on scale dependencies in claystone formations considering barrier properties such as porosity, permeability, and elastic properties in relation to burial history. The MATURITY project, launched in 2022, seeks to fill this gap through a field-to-lab-scale research initiative carried out within the Alamtheenton-Fm, a Lower Jurassic (Late Pliensbachian) organic matter-lean marine claystone. In this contribution, we set the outline for a series of detailed parameter studies designed to improve our understanding of burial-related changes in critical claystone properties for HLW disposal. We also provide an overview of the initial results.

The basis for the MATURITY project is a mineralogically homogeneous claystone formation with a natural maturity sequence attributed to variable burial and temperature history. This formation should be i) accessible with shallow drillings (<100 m below surface), and ii) covered by rock strata to minimize the influence of weathering processes. Mineralogical and sedimentological homogeneity is crucial as it ensures that observed changes in the investigated properties can be primarily attributed to variations in burial history rather than differences in depositional environment. The ideal scenario requires minimal natural variations in the depositional environment to maintain a mineralogically and geochemically homogeneous rock sequence throughout the study area. Correlations will be drawn between the analyzed properties and the burial history following a detailed study on key aspects of sedimentology and stratigraphy, geochemistry and mineralogy, petrophysics, geomechanics, and hydraulics of this claystone formation. These correlations will provide a particular emphasis for the applicability to potential site areas. The thermal maturity shall serve as key proxy for the burial history. Investigations on a field-to-lab scale shall enhance the understanding of spatial dependencies of the investigated properties, and further help to facilitate the data transition between the scale boundaries. Long-term observations with a focus on time-dependent quantification of transport properties shall be realized by establishing borehole-based investigation systems that allow permanent access to the investigated claystone formation.

To achieve the project objectives, we drilled eight shallow boreholes to a depth of approximately 100 m at five different sites that penetrate the claystones of the Amaltheenton-Fm at variable thermal maturities. These boreholes serve two main purposes: (a) to provide the necessary sample material (in the form of core samples) for extensive laboratory studies, and (b) to serve as access points for the investigations of multiple in-situ parameters relevant to HLW disposal. The general project approach is schematically shown in Fig. 2, and includes the vast majority of applied or planned methods. In this contribution, only selected data from different subdisciplines will be used to present a first characterization of the target formation, setting the project outline.



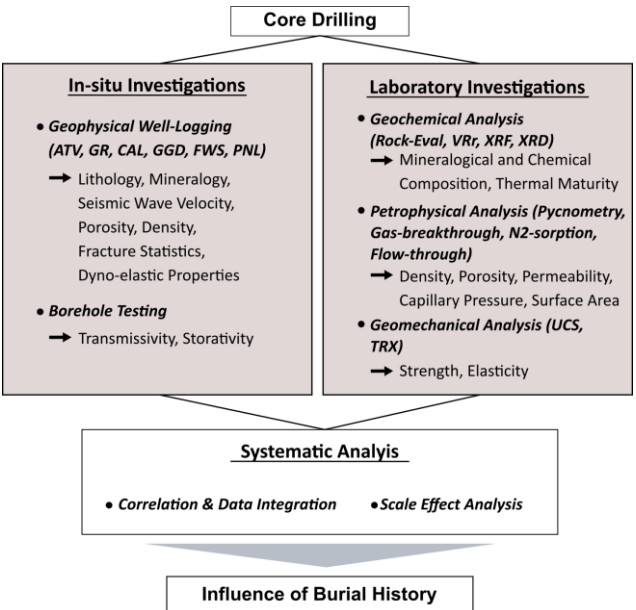

**Figure 2: MATURITY project overview with the principal project objectives and general investigation approach. ATV=Acoustic borehole televiewer, GR=Gamma-ray, CAL=Caliper log, GGD=Gamma-gamma-density, FWS=Fullwave Sonic log, PNL=Pulsed-neutron lifetime log, VRr=Vitrinite reflectance, XRF=X-ray fluorescence, XRD=X-ray diffraction, UCS=Uniaxial compressive strength test, TRX=Triaxial strength test.**

## 2 Site Identification

In Germany, the identification of a potential hostrock for the disposal of HLW is legally based on the Site Selection Act (StandAG, 2017). Based on the legal directives stipulated in the StandAG, the Federal Association for Radioactive Waste Disposal (BGE) was awarded the responsibility of identifying a site for HLW disposal that ensures safe long-term conditions (BGE, 2020). An overview of the methodological approach is given by Hoyer et al. (2021). This approach includes the division of Germany into several sub-areas in which geologically favorable conditions for HLW disposal are expected. In total, nine sub-areas cover claystone formations of Jurassic, Cretaceous, and Tertiary age (BGE, 2020, Fig.3).

For the MATURITY project, different maximum burial depths and temperatures of the target formation are essential in order to establish correlations between claystone properties and its burial history. Figure 3 illustrates the spatial distribution of sub-areas, as defined by BGE (2020), encompassing extensive regions of Northern Germany and parts of the federal states of Baden-Württemberg and Bavaria.

Within the Lower Saxony Basin (LSB), various authors have identified strongly variable thermal maturity of Lower Cretaceous and Jurassic rocks linked to deep burial during the Late Cretaceous (Bruns et al., 2016). Notable maturity variations were observed for the Toarcian Posidonienschiefer-Fm on a relatively small regional scale in southeastern Lower Saxony, as documented (among others) by Koch & Arnemann (1975), Littke et al. (1988), and Mackenzie et al. (1988). However, the





Posidonienschiefer-Fm itself is not regarded as a potential host rock for radioactive waste disposal due to its distinctive composition, characterized by a high carbonate and organic matter content, thus holding the potential for microbial activity,

enhanced chemical reactivity, and oil and gas generation (Rullkötter et al., 1988). In contrast, both the underlying Upper Pliensbachian Amaltheenton-Fm, and the overlying upper Toarcian to Aalenian sequences, comprising the Jurensismergel-Fm and Opalinuston-Fm, are being evaluated as discrete sub-areas for potential host rock suitability. Additionally, these claystones were uplifted to near surface levels by Cretaceous inversion tectonics, making them an ideal target horizon for shallow drillings within the scope of the MATURITY project. The Amaltheenton-Fm of the Upper Pliensbachian is finally selected because of

its accessibility and the presence of more than 100 m overlying clay- and marlstones.





**Figure 3: (a) Map of Mesozoic claystone sub-areas in northwestern Germany identified by BGE in accordance to §13 StandAG (modified after: BGE, 2021); (b) overview map of Lower Saxony with indication of the investigation area and isovitrinite reflectance contour lines (Isovitrinite reflectance modified after: Mackenzie et al., 1988; Klaver et al., 2012).**

## 170 2.1 Geological Outline

The Hils and the adjacent Sack Syncline structures are located in the southern part of Lower Saxony (Germany) approximately 50 km south of Hanover. Both are assigned to the southeastern margin area of the Lower Saxony Basin (LSB), which represents an E-W striking, highly differentiated Meso-Cenozoic basin system which is, in turn, part of the Central European Basin





System (CEBS) (Van Wees et al., 2000; Adriasola Muñoz, 2007; Maystrenko et al., 2008; Castro-Vera et al., 2024). The CEBS

formed initially as result of Late Carboniferous to Early Permian igneous activity, faulting and lithosphere subsidence (Van
Wees et al., 2000; Adriasola Muñoz, 2007; Burnaz et al., 2024). During this phase, the deposition of terrestrial sediments was
dominant. From the latest Permian to the Late Triassic the depositional setting was terrigenous to shallow-marine. A
progressive transgression from the onset of the Jurassic period led to marine sedimentation, which in the Early Jurassic
(Sinemurian to Toarcian) and the lower Middle Jurassic (Aalenian to Bajocian) resulted in the deposition of argillaceous

sequences with variable carbonate content. During the Toarcian, anoxic-euxinic conditions led to the formation of the organic
matter-rich Posidonienschiefer-Fm in a marginal marine environmental setting (Stahl, 1992; Maystrenko et al., 2008; Hooker
et al., 2020). During the Middle Jurassic repetitive sea level fluctuations (transgression-regression cycles) led to sandy
interlayers within marine claystones. The clastic influx ceased during the Callovian and the Oxfordian is characterized by open
marine claystone successions with interlayered carbonate sequences, especially in the upper Oxfordian (Stollhofen et al., 2008;

Bruns et al., 2013). Late Jurassic-Early Cretaceous rifting processes initiated the formation of the LSB, resulting in its rapid
subsidence while the adjacent blocks, e.g., the Rheinish Massif in the south and the Pompeckj Block in the north, experienced
strong uplift (Kockel et al., 1994; Senglaub et al., 2005). Sedimentation within the LSB remained predominantly shallow
marine with terrestrial sediments being confined to the transition between the uppermost Jurassic and lowermost Cretaceous
(Münder-Formation and Bückeberg-Group, Purbeck and Wealden Facies) (Stollhofen et al., 2008; Bruns et al., 2013).

Inversion tectonics, beginning in the Upper Cretaceous (Coniacian-Santonian) triggered the structural uplift of the LSB and
with it that of the Hils- and the Sack Syncline (Stahl, 1992; Voigt et al., 2021). Existing normal faults were reactivated and
transformed into steep thrust and reverse faults (Stahl, 1992; Petmecky et al., 1999; Baldschuhn & Kockel, 1998; Senglaub et
al., 2005). Later, the individual inversion structures underwent deep erosion beneath the Campanian unconformity and once
more before the Late Paleocene transgression (Stahl, 1992). Figure 4 summarizes the principal tecto-sedimentological

conditions and resulting geological structures and sequences. For more detailed information on the geodynamic and
sedimentological evolution of the CEBS and the LSB the reader is referred to (Betz et al., 1987; Van Wees et al., 2000;
Adriasola Muñoz, 2007; Bachmann et al., 2008; Bruns et al., 2013).

The lithostratigraphic terms used here correspond to the definition of the stratigraphic database Litholex compiled by the
German Stratigraphic Commission. Consequently, German terms are not translated and consistently used as proper terms

(litholex.bgr.de).





**Figure 4: Lithostratigraphic overview and geologic evolution of the LSB (modified after LBEG Stratigraphische Kommission, Castro-Vera et al., 2024).**

The Hils and Sack Synclines exhibit a lithological relief inversion towards the syncline center, with the Lower to Middle

Jurassic claystone sequences of the Pliensbachian to Aalenian dipping from the syncline margins towards the center at angles between 5° and 30° (Schmitz, 1980; Stahl, 1992; Wiese & Arp, 2013; Heunisch et al., 2018). The syncline centers are characterized by sequences of Upper Jurassic and Lower Cretaceous sedimentary rocks, primarily consisting of limestones, marlstones, and sandstones which form the geomorphological elevations of the Hils and Sack, and constitute part of the Weser-Leine Uplands. Wiese & Arp (2013) provide detailed litho-stratigraphic descriptions of the Hils- and Sack Syncline area. The

geological map and cross-section in Fig.5 outline the present-day local geological conditions of the Hils- and Sack syncline.



Isovitrinite lines, compiled from former studies of the region (Mackenzie et al., 1988; Castro-Vera et al., 2024) show the increasing thermal maturation trend from SE towards NW, and emphasize the strongly variable burial conditions within the investigated area.

**Figure 5: (a) Geological Map and (b) cross section of the Hils and Sack Syncline with indication of borehole locations from previous studies and the MATURITY project. Isovitrinite reflectance lines are added based on: Mackenzie et al. (1988); Castro-Vera et al. ( 2024). The geological cross section was modified after: Jordan et al. (1989); Wiese & Arp (2013).**



## 2.2 Previous Studies

Since the late 1980s, the area of the Hils Syncline has been investigated primarily focusing on the hydrocarbon generation
potential of the Posidonienschiefer-Fm, which represents western Europe´s most important petroleum source rock (Littke et
al., 1988; Rullkötter et al., 1988). Several shallow boreholes (appr. 60-80 m of depth) were drilled along the southern margin
of the Hils Syncline and penetrated a succession of Lower (Pliensbachian) to Middle (Aalenian) Jurassic claystones (Rullkötter
et al., 1988). Based on vitrinite reflectance values (VRr), the thermal maturation sequence increases from southeast to

northwest from 0.48 VRr% to 1.45 VRr% (Littke & Rullkötter, 1987; Littke et al., 1991). This trend has been repeatedly
reaffirmed by several studies (see Bernard et al., 2012; Klaver et al., 2012; Ghanizadeh et al., 2014). While early studies
assumed thermal maturation caused by a deep-seated mafic intrusion, younger studies (Petmecky et al., 1999; Senglaub et al.,
2005) more plausibly evidenced the maturity differences caused by differential burial. Recent investigations confirmed the
consistent depositional environments during the Pliensbachian (Burnaz et al., 2024) and local variations in burial and

temperature history (Fink et al., 2019; Castro-Vera et al., 2024) resulting in rather homogenous clay-dominated sequences with
different thermal maturities.

Mann (1987) documented a steady decrease in mean pore radii of Posidonienschiefer-Fm samples from 60 nm to 2.2 nm
between the Wenzen (Wen) and Harderode (Har) boreholes (Fig. 5). Slightly higher values of 3.2 nm were determined for
samples from the Haddessen (Had) borehole. Correspondingly, average porosities showed a gradual decrease with values

ranging from 20 % in Wenzen to partly <5 % in Harderode. A slight increase in porosity was observed between the Har and
the Had borehole, where porosities were measured at >8 %. The specific surface area was found to decrease from 18 to 0.5 m²/g
(Mann, 1987). These results were corroborated by similar findings in more recent studies, all showing a minimum porosity at
the intermediate maturity stage (oil-window, Harderode) (see Klaver et al., 2012; Gasparik et al., 2014; Ghanizadeh et al.,
2014; Rybacki et al., 2015; Grathoff et al., 2016; Mathia et al., 2016; Mohnhoff et al., 2016). The initial decrease in porosity

is interpreted as a combined effect of enhanced cementation and compaction during burial, while the increase from
intermediate-mature samples (0.91 VRr%) to post-mature samples (1.52 VRr%, gas-window, Haddessen) is seen as result of
secondary porosity evolution in the residual organic-matter (Klaver et al., 2012; Mathia et al., 2016; Mohnhoff et al., 2016).
Permeabilities were found to follow a similar trend with a range between $10^{-17}$ m² to $10^{-22}$ m² and a minimum permeability in
the intermediate maturity stage (Ghanizadeh et al., 2014; Grathoff et al., 2016; Mohnhoff et al., 2016). Rock mechanical tests

on specimens from the Toarcian formations did not yield systematic correlations between rock strength/elasticity and thermal
maturity (Rybacki et al., 2015). Mann & Müller, (1988) showed, among other findings, a trend of increasing density from
gamma-gamma (GGD) and neutron density data, corresponding to increasing thermal maturity quantified by the maximum
temperature during pyrolytic degradation ($T_{max}$).

However, in a pre-study towards the initiation of the MATURITY project, Gaus et al. (2022) investigated the petrophysical

(porostity, permeability) and mechanical (uniaxial compressive strength) characteristics on core materials from the 1980´s
drilling campaign. They were the first to focus on the organic-lean Amaltheenton-Fm. In their approach, thermal maturities,





derived from VRr values, were taken as proxy for burial depths and related temperatures, indicating maximum burial depths between 1,300 m and 3,600 m. Their results align with the previous studies on the organic-rich Posidonienschiefer-Fm, revealing an initial decrease of porosity and permeability until the intermediate maturation stage (or related burial depth of

approximately 2,650 m in Harderode) from 18 % to 5 % and 2.7 to 0.21 x $10^{-21}$ $m^2$, respectively. A slight increase in those properties was observed for over-mature samples from the Haddessen borehole (or related burial depth of 3,660 m). Uniaxial compressive strengths showed a reverse trend, increasing from 25 MPa to 40 MPa at burial depths ranging from 1.300 m to 2.650 m, whereas the sample from deeper burial of 3.660 m yielded a UCS value of 37 MPa. The latter observation contradicts the general burial trend. Possible explanations are given by Castro-Vera et al. (2024) who published a 3D basin model based

on data from newly drilled boreholes during the MATURITY project and the Wenzen borehole. Following this model, a gradual increase in maximum burial depth from 1,400 m (BO2.0) to 3,300 m (BO5.0) is responsible for stated trends in thermal maturity. However, localized overpressure generation due to undercompaction in the Amaltheenton-Fm and overpressure propagation originated in the overlying Posidonienschiefer-Fm due to gas generation likely explains the stated trends of decreasing density and increasing porosity within the Amalthea Clay Fm below 2,440 m.

## 265 3. Drillings, Sampling, and Borehole Installations

### 3.1 Borehole Selection and Drilling Processes

The basis for the proposed investigations on the Amaltheenton-Fm are six newly drilled boreholes within the western flank of the Hils Syncline and two boreholes on the south-eastern margin of the Sack Syncline. The selection of the borehole locations is based on (a) the findings of previous studies stating an increasing trend of thermal maturity from southeast to northwest, and

(b) structural field data (dip and dip direction) measured in outcrops and taken from old borehole data to estimate the upper boundary of the Amaltheenton-Fm, aiming to minimize the influence of surface weathering.

In total eight new boreholes at five different locations (BO1, BO2, BO3, BO4, BO5) were drilled in the scope of the MATURITY project from July 2022 to October 2023 within the target area (Fig. 5). Three locations (BO1, BO3, and BO5) were equipped with doublet boreholes with a horizontal spacing of approximately 5 m. All boreholes were fully cored in stable

rock sequences by means of a triple tube core barrel and a polymer-based drilling fluid (Pure-Bore©) to prevent clay mineral swelling. Unstable rock and soil sequences were stabilized by permanent, cemented metal casings. An overview of the individual borehole locations and specifications is given in Table 1.

**Table 1: Summary of borehole information. Masl=meters above sea level.**

| Location | Boreholes | Depth (m) | Elevation (masl) | Borehole Diameter (mm) |
|---|---|---|---|---|
| BO1 | BO1.0 | 92.5 | 252 | 146 |
| | BO1.1 | 94.0 | | |
| BO2 | BO2.0 | 99.0 | 227 | 146 |
| BO3 | BO3.0 | 102.0 | 156 | 146 |
| | BO3.1 | 101.0 | | |





| | BO4.0 | 95.0 | 168 | 146 |
|---|---|---|---|---|
| BO4 | BO5.0 | 98.8 | 119 | |
| BO5 | BO5.1 | 98.5 | | 146 |

## 3.2 Geophysical borehole investigations

Geophysical wire-line logging was carried out in the boreholes right after the drilling process. The applied logging methods comprise natural and spectral gamma-ray spectroscopy (NGR/SGR), acoustic borehole televiewer (ATV), resistivity, fullwave sonic (FWS), gamma-gamma-density (GGD), pulsed-neutron lifetime (PNL), temperature and caliper (CAL) logging. While GR, ATV, FWS, and CAL logs were recorded in all boreholes, GGD and PNL logging was only conducted in the boreholes BO1.1, BO3.0, and BO5.0. Because of borehole wall instability, borehole BO1.0 could only be logged with a temperature

probe and integrated natural gamma-ray tool while borehole BO2.0 could only be logged in an open borehole section between the casing (47 m) and a borehole blockage at 87 m of depth. In this contribution, selected logging results mainly from the Amaltheenton-Fm will be presented providing a first in-situ characterization of different formation properties.

Spectrometric Thorium/Potassium (Th/K) and Thoruium/Uranium (Th/U) ratios were derived from the SGR logging signals. The Th/K cross-plot, proposed by Quirein et al. (1982) and repetitively modified by Schlumberger (2009), is a widely used

technique in spectral gamma-ray log interpretation for identifying clay mineral assemblages (Rider, 2000; Klaja & Dudek, 2016). This method leverages the fact that different clay minerals have characteristic ratios of thorium to potassium due to their distinct geochemical nature and formation conditions, potentially providing knowledge of mineralogical composition (Weaver & Pollard, 1973; Fertl & Chilingarian, 1990; Rider, 2000; Klaja & Dudek, 2016; Gama & Schwark, 2022). In claystones a low Th/K ratio (Th/K < 4) can be associated with a dominance in illite, mica, glauconite, or feldspar minerals. In

contrast, higher Th/K ratios (Th/K > 6) may indicate an enrichment in kaolinite and/or heavy minerals (Rider, 2000; Gama & Schwark, 2022). The ratio between thorium and uranium, on the other hand, serves as an indicator of the depositional environmental, weathering, and redox conditions (Adams & Weaver, 1958; Klaja & Dudek, 2016). In a first approximation, we use these empirical correlations to give a general idea of the clay mineralogical composition of the Amaltheenton-Fm as well as compare the Amaltheenton-Fm at the individual locations. Bulk densities ($\rho_{bulk}$) were sampled from GGD logging

signals. Porosities were derived by applying:

$$\phi_{dyn} = \frac{\rho_{matrix} - \rho_{bulk}}{\rho_{matrix} - \rho_{fluid}} \tag{1}$$

with $\Phi_{dyn}$ being the in-situ porosity derived from logging signal. Porosity calculations were evaluated under the assumption of a constant matrix density ($\rho_{matrix}$) of 2.7 g/cm$^3$ and a fluid density ($\rho_{fluid}$) of 1.0 g/cm$^3$.

## 3.3 Hydrogeological borehole investigations

Hydraulic borehole tests were conducted using either a mobile straddle double packer system (BO2.0 and BO4.0) at various borehole depths or fixed Standpipe Double Packer Systems (SPDP) with one isolated hydraulic interval. A first test series of slug withdrawal (SW) tests was conducted in boreholes BO1.0, BO2.0, BO3.1, and BO4.0. For evaluation of Hydraulic





parameters (transmissivity, hydraulic conductivity) and corresponding flow dimensions, the pressure over time test data was analyzed using the n-dimensional Statistical Inverse Graphical Hydraulic Test Simulator, nSIGHTS (Roberts, 2006).

### 3.3.1 Borehole Installations

In-situ long-term hydraulic observations aiming at time-dependent quantification of flow and transport properties within the Amaltheenton-Fm are conducted by establishing borehole-based investigation systems. For this purpose, the doublet boreholes at the locations BO1, BO3, and BO5 were equipped with Standpipe Double Packer Systems (SPDP), schematically shown in Figure 6. The installed systems consist of:

-   Two inflatable straddle packers for the hydraulic isolation of borehole sections (Fig. 6c). Packers were inflated to 3 MPa inside the boreholes by injecting water. The interval between both packers is 5 m long and contains five 15-micron sinter metal filter elements of 1 m length each (Fig. 6c).
-   Stainless steel tubing pipes with space for two standpipes and an operation line for packer lines.
-   Two standpipes which provide access to the interval.
    -   o   **Standpipe 1** (SP1) allows the execution of hydraulic borehole tests (injection or withdrawal). SP1 is closed at the lower end with a pneumatic downhole valve.
    -   o   **Standpipe 2** (SP2) is equipped with a pressure-temperature sensor (P/T) at the lower end, constantly measuring the interval pressure and temperature. It is sealed with a mini-packer on top of the P/T sensor.
-   Caverns of 2 m depth and 1.2 m diameter, on the top of each borehole for data acquisition and system operation equipment (Fig. 6b,d) containing:
    -   o   Flowboard with manometers, valves, and pressure-line connections for pressure regulation of packers.
    -   o   Data logger for constant data acquisition.
    -   o   Nitrogen tanks for pressure stabilization of packers.
    -   o   Upper end of the steel tubing pipes with access to SP1 and SP2.

The systems installed allow hydraulic testing of isolated intervals through the injection or extraction of fluid into or from the interval. Continuous recording of pressure and temperature conditions via the P/T sensor installed in SP2 enables monitoring the long-term evolution of the interval pessure. Through numerical analysis, this data can be analyzed to derive information about permeability, storage capacity, and pore pressure conditions. Interference tests between the adjacent boreholes are designed to facilitate the quantification of these parameters beyond the borehole-influenced zone, and changes between fracture- and matrix-dominated flow. Furthermore, the extraction of water from the test interval (via SP1) provides the opportunity to sample and analyze the in-situ water. Equilibration of pressure conditions, until steady-state conditions were reached followed the installation and inflation of the SPDP system.



**Figure 6: (a) Schematic sketch of the Standpipe Double Packer System (SPDP) as installed within six boreholes of the Maturity-project, (b) data cavern with SPDP installations, (c) SPDP interval section with filter elements and packers as installed in six boreholes, (d) detail view on data cavern installations showing flowboard with 1. manometers and valves for packer-pressure regulation, 2. Data logger, 3. SPDP with openings of SP1, SP2 and operation line for packer regulation, and 4. Nitrogen tanks for regulation of packer pressures.**

## 3.3 Sampling and Sample Handling

Claystone drill core samples are prone to alteration processes after being retrieved from their in-situ conditions, potentially introducing irreversible changes in their properties (Ma & Zoback, 2018). Unloading due to the loss of the formations confining





pressure can lead to decompaction, volume expansion, and result in fracturing, such as the initiation of micro-fractures (Basu et al., 2020). The exposure to air leads to the loss of pore water, hence desaturation, shrinkage, and potential mineral precipitation, causing desiccation damage. In contrast, contact with brine can result in water uptake, dissolution and swelling

(Ewy, 2015). These alterations imply changes in the porosity, permeability, and mechanical characteristics of claystone drill core samples during and after retrieval. Consequently, appropriate drill core and sample handling is required to minimize alteration processes of the sample material, and consequent changes in petrophysical and mechanical properties of the claystones.

A protein-based polymer drilling additive (Pure-Bore ©) was used to prevent mixing of drilling fluid and formation water and

associated disequilibria that can lead to clay mineral swelling. Drill cores with a diameter of 101 mm were cored in 3-meter PVC liner intervals which were cut into 1-meter long segments. Samples of approx. 1 cm thickness were cut from the individual core end-faces right after core extraction for immediate geochemical testing. The core segments were closed with PVC end caps. Every second core segment was individually sealed in opaque aluminum foil, and de-aired to maintain a constant water content. For transport and storage, the cores were stored in wooden boxes. The tightness of the sealing was controlled on a

regular basis, and renewed if necessary.

### 3.4 Laboratory Investigations

A comprehensive, multidisciplinary laboratory program is currently being conducted on core material from the boreholes. This program encompasses methods for geochemical characterization of the Amaltheenton-Fm, petrophysical investigations, as well as mechanical laboratory tests. In this contribution, the presentation and analysis of data focuses on selected methods,

aiming for a first characterization of the Amaltheenton-Fm. Results of other lithologies and formations are concisely presented. A comprehensive analysis of the full set of laboratory and borehole data is outside the scope of this contribution and will be addressed in individual parameter studies.

Core samples from boreholes BO1.0 - BO5.0 were continuously sampled at intervals ranging from 0.5 to 1 meter for geochemical analysis. Rock-Eval pyrolysis was employed using a Rock-Eval 7 device by Vicini Technologies to determine

already generated, free hydrocarbon S1 (mg HC/ g rock), the hydrocarbon potential S2 (mg HC/g rock), and generated $CO_2$ S3 (mg CO2/g rock). Total Organic Carbon (TOC) normalized Hydrogen Index (HI) and Oxygen Index (OI) were calculated. Additionally, the organic (TOC) and inorganic carbon content (TIC, and related carbonate content) were derived based on the pyrolysis data. The temperature of maximum hydrocarbon generation ($T_{max}$) was determined and corrected to Rock-Eval standards. For a detailed methodology of the applied measurements see Burnaz et al. (2024) and references therein.

Quantitative X-ray diffraction (XRD) analysis was conducted on powdered bulk materials to obtain the mineralogical composition of the target formation. Gently crushed bulk materials were spiked with 0.2 g/g internal standard ($\alpha$-$Al_2O_3$) for accuracy control and ground in a McCrone Micronizing mill adding ethanol as cooling lubricant to prevent heat- and



mechanical strain damage. The air-dried powders were measured on a Bruker D8 diffractometer using Cu Kα radiation (40 kV, 40 mA). The mineralogical composition was quantified based on Rietveld refinement employing the BGMN-based software Profex in combination with customised clay-structure models (Ufer et al., 2008; Doebelin & Kleeberg, 2015; Ufer & Kleeberg, 2015). The major elemental composition was obtained from X-Ray fluorescence. Powder pellets prepared from 8 g of powdered bulk material (<63 μm) combined with 2 g of Fluxana CEREOX wax and pressed for 120 seconds at 19.2 MPa. The analysis was performed on an energy dispersive polarised SPECTRO XEPOS ED(P)-X-Ray fluorescence instrument (limit of detection ≤1.4 μg/kg) (Grohmann et al., 2023; Spectro, 2007). Every sample was measured twice, rotating the pellet in between. Subsequently, mean values were calculated for the two measurements. In this contribution, the presentation of the elemental analysis results focuses on aluminum, silicium and iron.

Porosities were calculated on several specimens from intact core sections of the boreholes BO1.0 - BO5.0. Bulk densities and skeletal densities for porosity determination were derived from caliper and weight measurements, and he-pycnometry, respectively, following the methods described by Gaus et al. (2022) and references therein.

Permeability was measured under unconfined conditions using a modified pycnometry-based method. The applied modifications build upon concepts and experimental techniques reported by Li et al. (2006), Gaus et al. (2019), and Khajooie et al. (2025). In this approach, permeability is measured via radial gas uptake by sealing the axial surfaces of the core sections using epoxy. The measurements were then evaluated based on mathematical solutions for diffusion in a cylinder in the radial direction, along with a diffusivity equation that relates diffusion to permeability and porosity (Crank, 1975; Li et al., 2006; Khajooie et al., 2025). Nitrogen was used as the permeating gas.

## 4. Site Characterization

### 4.1 Lithological borehole Sequences

Lithological borehole interpretations were derived from the geophysical logging data and from geochemical analysis of the extracted core material. Lithological borehole profiles of selected boreholes are presented in Fig. 7 alongside the respective spectral- (SGR) and computed GR (CGR) signals.



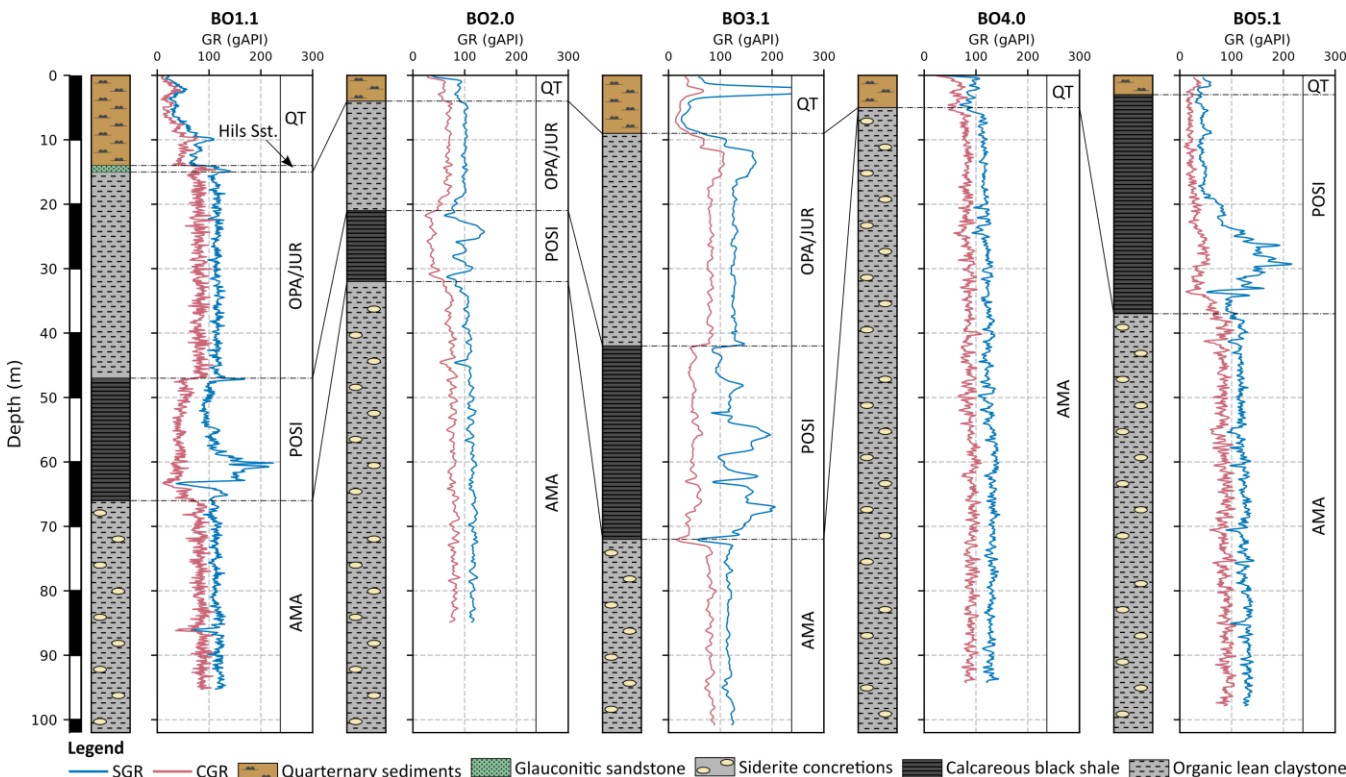

**Figure 7: Lithostratigraphic correlations of selected boreholes from each Maturity-project drilling location with gamma-ray logging signals (SGR=Spectral Gamma-Ray; CGR=Computed Gamma-Ray). Qt.=Quaternary, Sst=Sandstone, JUR=Jurensismergel-Fm, OPA=Opalinuston-Fm, POSI=Posidonienschiefer-Fm, AMA=Amaltheenton-Fm.**

At location BO1, the first 14 m of the boreholes penetrated Quaternary unconsolidated sediments. Between 14 m and 15 m depth, a peak in GR-readings indicates a layer of Cretaceous glauconitic sandstones, which can be identified as part of the Lower Cretaceous Hils-Formation. From 15 m depth onwards, grayish claystones (Opalinuston-Fm/Jurensismergel-Fm) extend to a depth of approximately 46 m. At this point, a peak in GR-signal (168 API) suggests the transition into the Posidonienschiefer-Fm. The Amaltheenton-Fm follows from a depth of 65 m and constitutes the borehole lithology down to

the final depth of 98 m. The boreholes at locations Mainzholzen, and Hunzen penetrated a succession of Quaternary sediments followed by the Opalinuston-Fm/Jurensismergel-Fm, and the Posidonienschiefer-Fm, overlying the Amaltheenton-Fm. In BO4.0, both the Opalinuston-Fm/Jurensismergel-Fm and Posidonienschiefer-Fm are entirely absent. Below approximately 6 m of unconsolidated Quaternary/Tertiary sediments, the Amaltheenton-Fm was encountered and constitutes the entire borehole lithology to the final depth of 95 m. The boreholes at the Bensen location (BO5.0 and BO5.1) penetrated the

Posidonienschiefer-Fm below a few meters of Quaternary/Tertiary overburden, while the Opalinuston-Fm/Jurensismergel-Fm is absent. The Amaltheenton-Fm was encountered at a depth of 37 m (BO5.0) and 38 m (BO5.1), respectively.

Where existent, the Posidonienschiefer-Fm stands out in GR logging signals as a distinct peak with high GR readings (>140 API) due to its high organic-matter content (mean TOC from 6.4 wt.% to 9.7 wt.%) and a correspondingly high uranium



concentration. The Posidonienschiefer-Fm serves as marker horizon for the stratigraphic classification of the boreholes. Its
pronounced geophysical and geochemical signatures make it a valuable tool for the regional correlation and stratigraphic
interpretation between the individual borehole locations. The transition to the Amaltheenton-Fm is indicated by a decrease in
the gamma-ray log response. The Amaltheenton-Fm is characterized by constant GR readings between 110-120 API. Here,
little variation in the logging signal indicates a homogeneous composition between the individual locations. In the extracted
drill cores, the Amaltheenton-Fm generally appears as a gray, fine bedded claystone with a frequent appearance of yellow-
brownish carbonate concretions (Fig. 8) identified as siderite concretions. The occurence of siderite concretions varies from
millimeter- to centimeter- long spheroidal nodules to decimeter-thick sideritic "lumps".

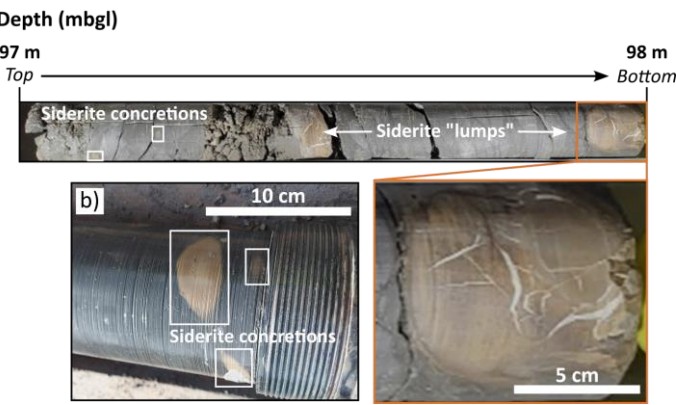

**Figure 8: Exemplary rock core sections of the Amaltheenton-Fm with emphasis on siderite concretions of variable size and shape:**
**(a) Drill core section from borehole BO2.0 between 97 m and 98 m below ground level (mbgl), and (b) core section from borehole**
**BO4.0 with indication of siderite nodules.**

### 4.2 Geochemical and Mineralogical Characteristics

Spectrometric ratios of the elements Th, K, and U were derived from the SGR logging signals. Little variation and consistent
clustering of Th/K ratios (Fig. 9a) can be observed in the Amaltheenton-Fm between the individual test sites, suggesting a
homogenous clay mineral assemblage throughout the formation. The Th/K cross plot suggests a clay mineralogy dominated
by mixed-layer clay with mean Th/K ratios between 7.1 (BO1.1) and 7.98 (BO3.0). Mean Th/U ratios range between 4.87
(BO2.0) and 7.04 (BO3.0). In BO1.1 Th/U ratios of 6.82 are almost equally high as those in BO3.0, whereas BO4.0, and BO5.0
show values closer to BO2.0, at 5.84 and 4.94, respectively. However, generally, the Th/U ratios show overlapping clustering
as indicated in Fig. 9b.



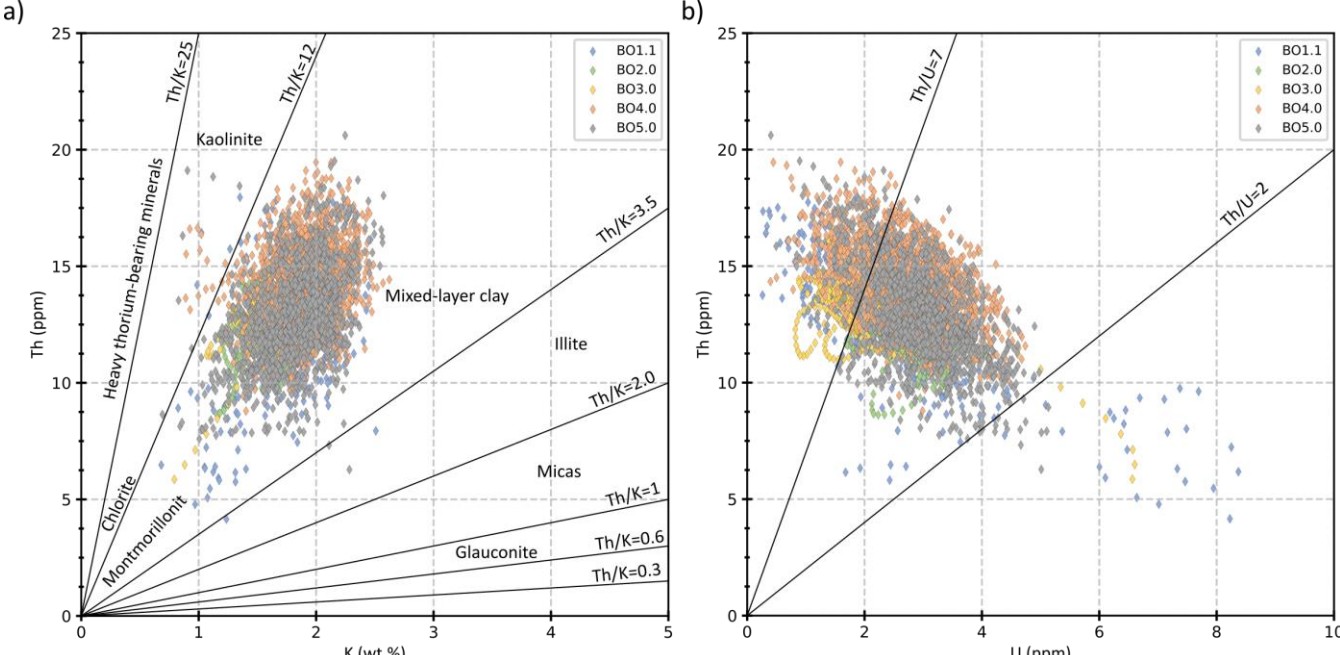

**Figure 9: Elemental ratios of the Amaltheenton-Fm from SGR logging signals: (a) Thorium versus potassium mineral identification plot from Schlumberger (2009) with plotted SGR readings from the boreholes BO1.1, BO2.0, BO3.0, BO4.0, BO5.0; (b) Thorium versus uranium plot.**

Quantitative X-ray diffraction (XRD) and X-ray fluorescence (XRF) analyses were conducted to determine the mineralogical and geochemical composition of the Amaltheenton-Fm. The results characterize the Amaltheenton-Fm as a claystone dominated by high clay contents ranging between 58 wt.% (BO1.0) and 75 wt.% (BO2.0) (Fig. 10). The framework silicates Quartz + Feldspars contribute between 15 wt.% to 23 wt.%, and 7 wt.% to 12 wt.%, respectively. The carbonate content within the Amaltheenton-Fm is generally low with a maximum concentration of 8 wt.% in borehole BO5.0. Exceptionally high but spatially confined carbonate contents of up to 80 wt.% were observed for several carbonate concretions, i.e. siderite, across all locations in the Amaltheenton-Fm. Quantitative XRD analyses revealed siderite as primary component of these samples, endorsed by increased Fe values from XRF analysis (Fig. 10b). Siderite concretions occur in the Amaltheenton-Fm in all boreholes but were not sampled to the same extent, resulting in an apparent siderite abundance in boreholes BO2.0 and BO4.0 in Fig.10. According to the classification scheme provided by Lazar et al. (2015), the Amaltheenton-Fm can be defined as argillaceous fine-grained sedimentary rock. Table 2 provides a summary of the general mineralogical composition of the Amaltheenton-Fm. The results align with findings from Gaus et al. (2022) on sample material from the old core material.

**Table 2: Summary of mean XRD derived mineralogical composition of the Amaltheenton-Fm. Standard deviations are given beside the mean values. Sample populations are indicated for each borehole with (#).**

| Borehole | Quartz (wt.%) | Clays (wt.%) | Carbonates (wt.%) | Feldspars (wt.%) | Accessories (wt.%) |
|---|---|---|---|---|---|
| BO1.0 (#9) | 19.94 ± 3.34 | 64.63 ± 4.76 | 2.47 ± 2.49 | 8.62 ± 0.46 | 4.34 ± 1.04 |





| | | | | | |
|---|---|---|---|---|---|
| BO2.0 (#6) | 14.58 ± 1.80 | 75.69 ± 1.67 | 1.17 ± 0.65 | 7.41 ± 0.26 | 1.14 ± 0.47 |
| BO3.0 (#4) | 17.15 ± 2.20 | 59.86 ± 1.76 | 4.82 ± 2.55 | 11.81 ± 1.38 | 4.11 ± 1.87 |
| BO4.0 (#8) | 19.35 ± 3.48 | 63.11 ± 2.67 | 5.59 ± 4.96 | 8.75 ± 2.83 | 3.28 ± 0.59 |
| BO5.0 (#6) | 22.84 ± 6.66 | 58.40 ± 5.46 | 8.00 ± 2.42 | 6.96 ± 0.96 | 3.80 ± 0.60 |

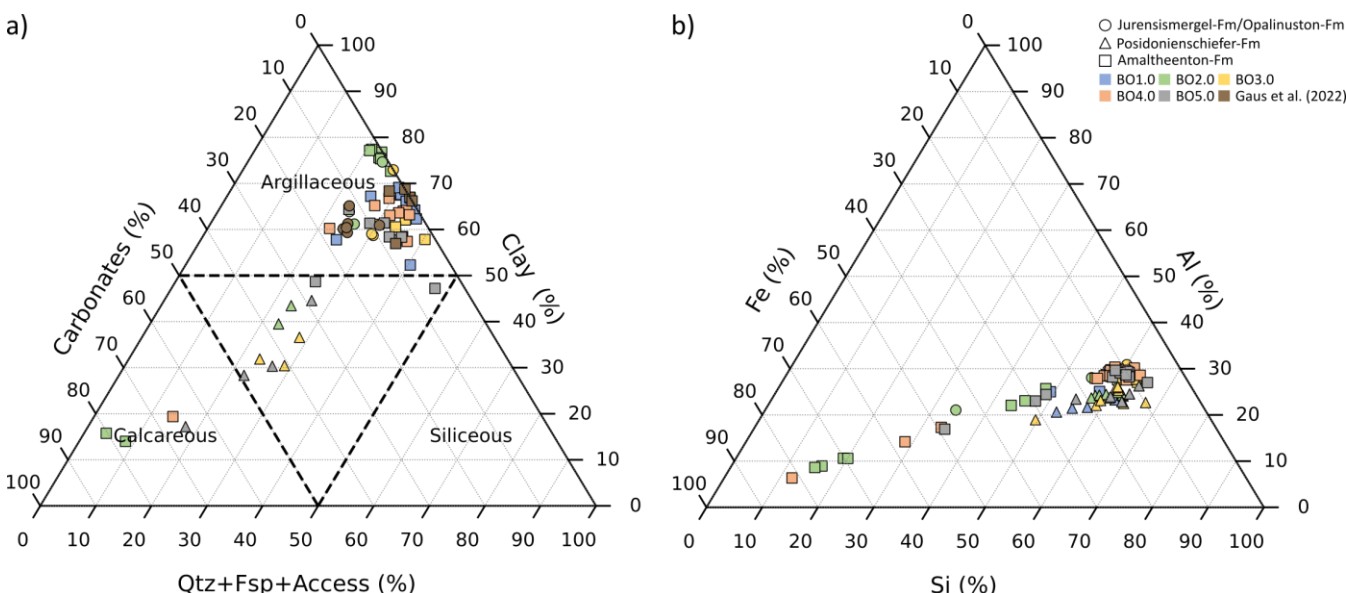

**Figure 10: Ternary plots of (a) mineralogical composition from XRD analysis with data from the Maturity-project boreholes and old boreholes after Gaus et al. (2022). Nomenclature guidelines for the composition of fine-grained sedimentary rocks are added based on Lazar et al. (2015); (b) the Fe-Si-Al phase from XRF analysis of boreholes BO1.0, BO2.0, BO3.0, BO4.0, and BO5.0.**

## 4.2 Thermal Maturation Sequence

The vitrinite reflectance was determined on selected samples obtained from core sections from five of the boreholes (BO1.0 - BO5.0). Most of these samples are extracted from the Amaltheenton-Fm borehole intervals, since the Posidonienschiefer-Fm is depleted in vitrinite (Littke et al., 1988). Vitrinite reflectance in boreholes BO1.0 and BO2.0 is similar at about 0.52% and 0.48% VRr, respectively, while higher values are recorded for boreholes BO3.0, BO4.0, and BO5.0 (0.70%, 0.87%, 1.51% VRr, respectively). These values agree well with those published earlier for adjacent boreholes on the southwestern flank of the Hils Syncline and with values from numerous Jurassic outcrop samples (Castro-Vera et al., 2024; Littke et al., 1988).

Further information on thermal maturity is provided by elemental analysis and Rock-Eval data. Average Rock-Eval data for the drilled formations is summarized in Table 3. TOC and carbonate contents are high in the Posidonienschiefer-Fm, moderate in the Jurensismergel-Fm/Opalinu Clay Fm, and low in the Amaltheenton-Fm. In Fig. 11 Rock-Eval derived data from BO2.0 is plotted against depth to exemplarily visualize this trend.



**Figure 11: Bulk geochemical data of BO2.0. Total organic carbon (TOC), CaCO₃, and hydrogen index (HI) data were derived from Rock-Eval pyrolysis. In addition, spectral- and computed gamma-ray (SGR, and CGR, respectively) are given.**

Carbonate content is not significantly affected by thermal maturity as visible from a comparison of boreholes BO1.0 and BO5.0, while TOC values decrease from the immature to the overmature state, e.g. in the Posidonienschiefer-Fm from about 10 % to about 6 %. This observation is in accordance with the mass balance results published in Rullkötter et al. (1988). Furthermore, Hydrogen Index values are very high in the Posidonienschiefer-Fm, moderate in the Jurensismergel-Fm/ Opalinuston-Fm and rather low in the Amaltheenton-Fm. However, during thermal maturation, the Hydrogen Index is much reduced due to petroleum generation and expulsion; therefore, values in the Posidonienschiefer-Fm are much lower in BO5.0 (Table 3). This trend is also observed for the Amaltheenton-Fm, but it is less obvious, because initial values in, e.g., borehole BO1.0 are already low.





Concerning these thermal maturity comparisons, it must be emphasized that the same formation should always be compared. This is done here, but not always the same stratigraphic range has been drilled. An almost complete profile of Posidonienschiefer-Fm only exists for BO3.0 and BO5.0, and the Amaltheenton-Fm was drilled in thicknesses ranging from less than 30 m to almost 100 m.

**Table 3: Summary of organic and inorganic carbon content (TOC, TIC), calcite content (CaCO₃), hydrogen index (HI), oxygen**
**index (OI), peak S2 temperature (T$_{max}$), and vitrinite reflectance for the Amaltheenton-Fm from boreholes BO1.0 to BO5.0. Sample populations are indicated for each borehole with (#).**

| Borehole | Formation | TOC ± Std. (wt.%) | TIC ± Std. (wt.%) | CaCO₃ (wt.%) | HI ± Std. (mg HC/gTOC) | OI ± Std. (mg CO₂/g TOC) | Tmax ± Std. (°C) | VRr ± Std. (VRr%) |
|---|---|---|---|---|---|---|---|---|
| | JM/OPA (#29) | 2.10 ± 0.33 | 0.85 ± 0.58 | 7.06 | 225 ± 39 | 21 ± 7 | 434 ± 3 | |
| BO1.0 | POSI (#17) | 9.73 ± 3.90 | 4.42 ± 2.92 | 36.69 | 6845 ± 49 | 7 ± 1 | 418 ± 5 | |
| | AMA (#27) | 0.84 ± 0.10 | 0.20 ± 0.13 | 1.66 | 128 ± 38 | 39 ± 15 | 437 ± 4 | 0.52 ± 0.04 (#5) |
| | JM/OPA (#17) | 1.23 ± 0.11 | 1.50 ± 0.25 | 12.45 | 242 ± 30 | 32 ± 6 | 430 ± 3 | |
| BO2.0 | POSI (#10) | 9.63 ± 3.91 | 4.28 ± 0.68 | 35.52 | 700 ± 210 | 16 ± 25 | 422 ± 1 | |
| | AMA (#59) | 0.88 ± 0.14 | 0.37 ± 0.22 | 3.07 | 76 ± 27 | 34 ± 12 | 431 ± 3 | 0.48 ± 0.04 (#5) |
| | JM/OPA (#24) | 1.71 ± 0.40 | 0.84 ± 0.29 | 6.97 | 294 ± 103 | 24 ± 25 | 435 ± 4 | |
| BO3.0 | POSI (#9) | 8.56 ± 1.01 | 5.80 ± 0.54 | 48.14 | 652 ± 28 | 4 ± 1 | 441 ± 2 | |
| | AMA (#10) | 0.74 ± 0.07 | 0.37 ± 0.14 | 3.07 | 151 ± 31 | 38 ± 12 | 444 ± 3 | 0.70 ± 0.05 (#5) |
| BO4.0 | AMA (#65) | 0.83 ± 0.11 | 0.43 ± 0.51 | 3.57 | 78 ± 22 | 40 ± 22 | 447 ± 2 | 0.87 ± 0.08 (#5) |
| BO5.0 | POSI (#21) | 6.40 ± 1.15 | 4.80 ± 140 | 39.84 | 60 ± 6 | 4 ± 1 | 463 ± 5 | |
| | AMA (#52) | 0.81 ± 0.16 | 0.38 ± 0.22 | 3.15 | 25 ± 13 | 65 ± 61 | 462 ± 28 | 1.51 ± 0.08 (#10) |

Aside from HI values, Rock-Eval T$_{max}$ values are dependent on thermal maturity. In the Posidonienschiefer-Fm, T$_{max}$ values increase from BO1.0/BO2.0 (about 420 °C) to BO3.0 (about 441 °C) and BO5.0 (463 °C). A common plot on thermal maturity is the HI (Hydrogen Index) versus T$_{max}$ diagram (Fig. 12). Standard lines separating between immature, mature, and
overmature conditions for petroleum generation (0.5 VRr%, and 1.2 VRr%, respectively) are additionally plotted. However, these lines should be considered as a rough orientation. The progress in thermal maturity is obvious from the plotted data. For BO4.0 only Amaltheenton-Fm data is available. However, BO4.0 is adjacent to the old Harderode borehole (Har), which penetrated the Posidonienschiefer-Fm. The respective data published by Rullkötter et al. (1988) fits also well into this trend. For BO5.0, only Posidonienschiefer-Fm samples can be used, because pyrolysis peaks for the Amaltheenton-Fm are too small
to evaluate T$_{max}$ values (see also very low HI values; Table 3).



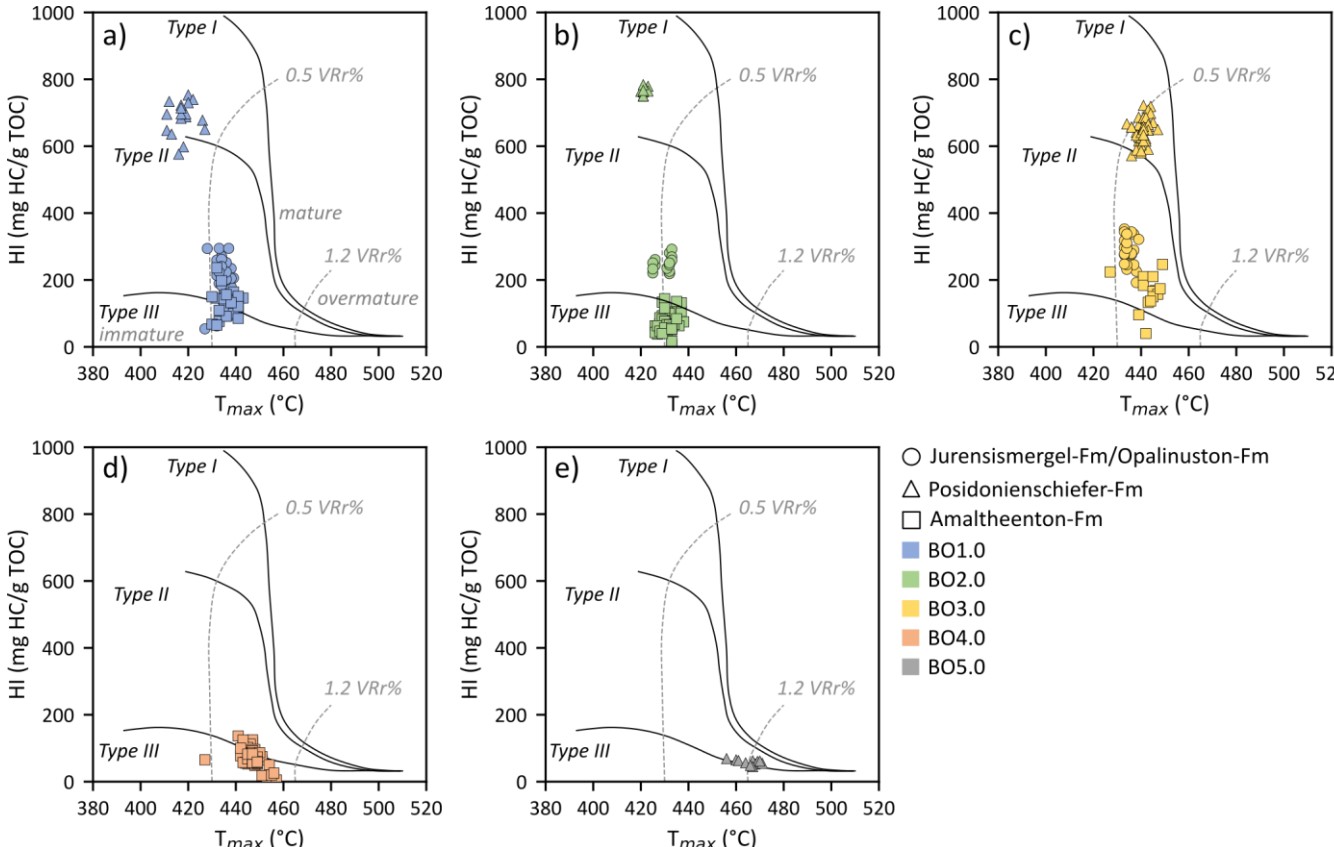

**Figure 12: Hydrogen Index vs. Tmax crossplots outlining the kerogen types I, II, and III and the thermal maturity for samples of Jurensismergel-Fm (JM)/Opalinuston-Fm (OC), Posidonienschiefer-Fm (Posi), and the Amaltheenton-Fm (AMA) from the boreholes drilled in the MATURITY project: (a) BO1.0 (blue), (b) BO2.0 (green), (c) BO3.0 (yellow), (d) BO4.0 (orange), and (e) BO5.0 (grey).**

## 4.3 Petrophysical and Hydrogeological Characteristics

Different geophysical logging methods were applied to assess in-situ petrophysical properties. Mean compressional wave velocities (Vp) from the Amaltheenton-Fm show an increasing trend from BO1.1/BO2.0 (2,367 m/s and 2,302 m/s) to BO4.0 (3,050 m/s), while a decrease is observed between BO4.0 and both boreholes at the borehole location Bensen (BO5.0 and BO5.1 with 2,555 m/s and 2,538 m/s, respectively see Fig. 13a). Gamma-density logs (GGD) from BO1.1, BO3.0, and BO5.1 are additionally presented in Fig. 13. Mean densities from the logging data of the Amaltheenton-Fm sections within those three boreholes are 2.45 g/cm$^3$, 2.51 g/cm$^3$, and 2.50 g/cm$^3$. Mean porosities were derived at 14.57 %, 11.26 %, and 11.56 %, respectively. In laboratory measurements, mean densities range between 2.31 g/cm$^3$ (BO1.0) and 2.49 g/cm$^3$ (BO4.0), while porosities derived from He-pycnometry are between 8.96 % and 13.58 % in BO4.0 and BO2.0, respectively. A similar trend to Vp velocities is observed in the lab density data, showing an increase in density between the borehole locations BO1.0 and BO4.0, whereas densities in BO5.0 average slightly below those from BO4.0 with 2.46 g/cm$^3$. Conversely, porosities initially



decrease in the order BO2.0>BO1.0>BO3.0>BO4.0, but show an increase between BO4.0 (8.96 %) and BO5.0 (10.26 %).
Permeability data from unconfined conditions further reflects variations seen in porosities. Between BO2.0 and BO4.0, the
permeability decreases from $1.48 \times 10^{-19}$ m$^2$ to $1.25 \times 10^{-20}$ m$^2$ then slightly increases to $1.61 \times 10^{-20}$ m$^2$. However, no significant
differences were observed between BO1.0 and B02.0. Table 4 summarizes log- and lab-derived petrophysical data from the
Amaltheenton-Fm.

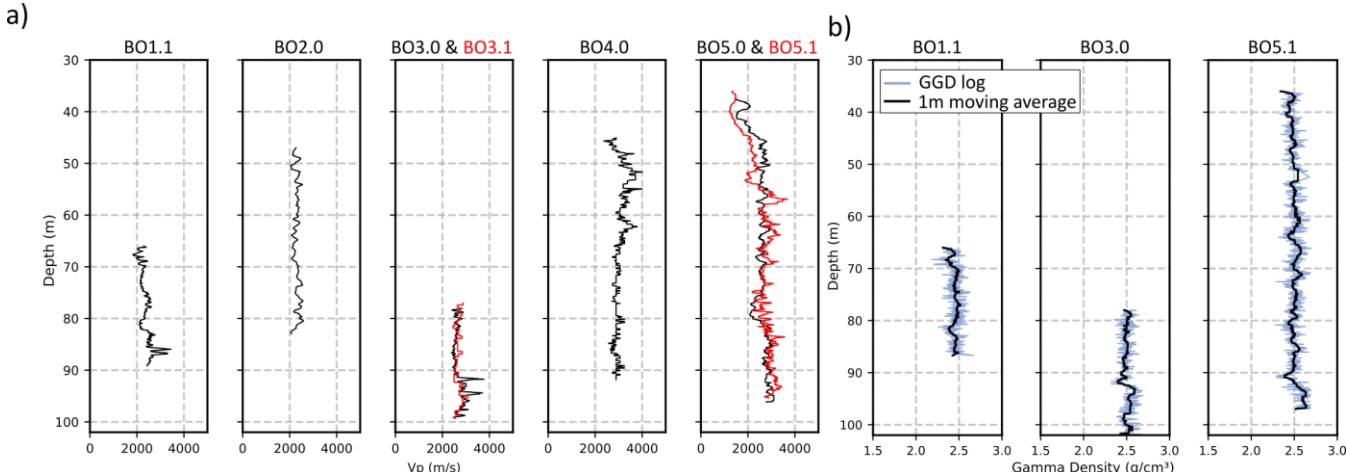

**Figure 13: Logging sections in the Amaltheenton-Fm: (a) Compressional wave velocities (Vp) from sonic logs, and (b) gamma-density logs (GGD) of the boreholes BO1.1, BO3.0, and BO5.1.**

**Table 4: Basic petrophysical rock data of the Amaltheenton-Fm derived from logging and laboratory data. Log densities and porosities were derived from gamma-density logs. Laboratory data was determined via He-pycnometry and caliper measurements.**

| Borehole | Mean Vp (m/s) | Gamma-Density (g/cm$^3$) | Gamma-Porosity (%) | Density (g/cm$^3$) | Porosity (%) | Permeability (k·10$^{-19}$m$^2$) | Equivalent hydraulic conductivity (K·10$^{-12}$m/s) |
|---|---|---|---|---|---|---|---|
| BO1.0 | - | - | - | 2.31 | 12.47 | 1.47 | 1.81 |
| BO1.1 | 2367 | 2.45 | 14.57 | - | - | - | - |
| BO2.0 | 2301 | - | - | 2.35 | 13.7 | 1.48 | 1.81 |
| BO3.0 | 2702 | 2.51 | 11.26 | 2.45 | 9.15 | 0.89 | 1.09 |
| BO3.1 | 2681 | - | - | - | - | - | - |
| BO4.0 | 3050 | - | - | 2.48 | 8.96 | 0.13 | 0.15 |
| BO5.0 | 2555 | - | - | 2.46 | 10.26 | 0.16 | 0.20 |
| BO5.1 | 2538 | 2.50 | 11.56 | - | - | - | - |

A first series of hydraulic tests from boreholes BO1.0, BO2.0, BO3.1, and BO4.0 was conducted and evaluated. The results
are presented in Table 5. Hydraulic conductivities span two orders of magnitude ranging from $1.15 \times 10^{-7}$ m/s (BO4.0) to
$2.68 \times 10^{-5}$ m/s.




**Table 5: Summary of hydraulic test results. Presented data was evaluated from pulse withdrawal (PW) and slug withdrawal tests (SW). T=transmissivity, K=hydraulic conductivity, n=flow dimension.**

| Borehole | Test Type | Investigation Depth (m) | Interval Length (m) | Static Pressure (kPa) | Minimum Pressure (kPa) | T (m²/s) | K (m/s) | n |
|---|---|---|---|---|---|---|---|---|
| BO1.0 | SW | 78.50-82.84 | 4.34 | 609.0 | 342.0 | $3.12 \times 10^{-5}$ | $2.68 \times 10^{-5}$ | 1.4 |
| BO2.0 | SW | 63.0-70.50 | 7.50 | 631.19 | 543.61 | $1.91 \times 10^{-5}$ | $1.62 \times 10^{-6}$ | 1.7 |
| BO3.1 | PW | 92.0-96.34 | 4.34 | 454.52 | 10.76 | $3.02 \times 10^{-5}$ | $6.95 \times 10^{-6}$ | 2.9 |
| BO4.0 | SW | 31.90-39.00 | 7.10 | 339.65 | 225.0 | $8.17 \times 10^{-7}$ | $1.15 \times 10^{-7}$ | 2.3 |

## 5. Implications and Outlook

### 5.1 The Amaltheenton-Fm in regard to the MATURITY project

X-ray diffraction (XRD) and X-ray fluorescence (XRF) analyses conducted on Amaltheenton-Fm samples from all boreholes have revealed minor variability in mineralogical composition. Clay minerals constitute the dominant mineral type (58-75 wt.%) defining the Amaltheenton-Fm as argillaceous claystone. Th/K ratios derived from gamma-ray logs suggest a mixed-layered clay mineralogy that exhibits only slight variations between the individual borehole locations. Carbon data (TOC, TIC) also demonstrate low variance within the Amaltheenton-Fm. However, mean Th/U ratios of the Amaltheenton-Fm show some slight variability ranging between a factor of approximately 5 in boreholes BO2.0, BO4.0, and BO5.0, and a factor of approximately 7 in boreholes BO1.1 and BO3.0, which might be attributed to various reasons. Th/U ratios between 2 and 7 are associated with a marine depositional environment and moderate terrigenous supply, while values >7 indicate continental, oxidizing conditions, and values <2 marine, reducing conditions (Adams & Weaver, 1958; Rider, 2000; Klaja & Dudek, 2016; Bataller et al., 2022). A detailed paleo-environmental study on samples from boreholes BO2.0 and the adjacent borehole Wickensen (WIC) by Burnaz et al. ( 2024) revealed a shallow marine depositional setting with effective circulation under predominately oxic bottom water conditions and terrigenous clastic input for the Amaltheenton-Fm. Generally, this corresponds to the detected Th/U ratios. However, higher Th/U ratios, close to 7, as seen in BO1.1 and BO3.0 might be due to different stratigraphic ranges within the Amaltheenton-Fm and slightly divergent environmental conditions during deposition, characterized by stronger continental influence. Differences in Th/U ratios are also likely to be a result of secondary uranium mobilization and fluid migration from the overlying Posidonienschiefer-Fm, resulting in lower values at locations BO2.0, BO4.0, and BO5.0.

The increase in thermal maturity is documented by several parameters (VRr, TOC, HI, $T_{max}$). The maturity trend that was established for the Hils Syncline follows published data (see Littke et al., 1988; Castro-Vera et al., 2024), where SE-NW directed thermal maturity increase is interpreted as related to deep burial. Measured vitrinite reflectance values can be converted into maximum burial temperatures based on an empirical equation published by Barker & Pawlewicz (1994):



$$Tpeak\ (burial)\ =\ (ln(VRr)\ +\ 1.68)/0.0124 \tag{2}$$

When applied to BO1.0, BO2.0, BO3.0, BO4.0, and BO5.0 (with VRr% values of approximately 0.52, 0.48, 0.70, 0.87, and 1.51, respectively), the resulting maximum burial temperatures are 83°C, 76°C, 107°C, 124°C, and 169°C. Maximum burial depths can be established from $T_{peak}$ incorporating the sediment water interface temperature (SWIT) and the geothermal gradient at the time of maximum burial. Based on the 3D model provided by Castro-Vera et al. (2024) the SWIT temperature during deepest burial (latest Early Cretaceous) is sampled at approx. 24°C (personal communication with L. Castro-Vera) while a geothermal gradient of 30°C was assumed. Maximum burial temperatures and depths for the Amaltheemton-Fm from all investigated locations are given in Table 6. Note that $T_{peak}$ values established via Eq.2 and related burial depths reached over geologic times are not directly related to $T_{max}$ values obtained from Rock-Eval pyrolysis measurements. According to the temperatures calculated via Eq. 2, BO1.0 and BO2.0 are immature, BO3.0 and BO4.0 have reached the oil generation stage (oil window), and BO5.0 is in the gas generation stage. This fits well with the results from Rock-Eval pyrolysis. It should be noted that this temperature and burial depth calculation via Eq.2 does not consider the exact burial and temperature history but it is a simple, yet useful approximation. Short periods at the maximum temperature, e.g., during igneous intrusions, would lead to higher temperatures. One important aspect is attributed to the subtle difference between boreholes BO1.0 and BO2.0. The initial expectation was that BO1.0 situated outside of the Hils Syncline close to the southern tip of Sack Syncline might be thermally less mature than BO2.0. This is not the case according to the collected data; thermal maturity is very similar and $T_{max}$ values even tend to be slightly higher at BO1.0 than at BO2.0. Such subtle differences in the thermal maturity of immature rocks can well be investigated by biomarker studies in the future.

**Table 6 – Estimated maximum burial temperatures (Tpeak buria) and depths from vitrinite reflectance for the boreholes BO1.0 – BO5.0. 3D modelled maximum burial depth adapted from Castro-Vera et al. (2024) and modelled Tpeak values based on personal communication with L. Castro-Vera.**

| Borehole | Vitrinite reflectance (VRr%) | Tpeak burial (°C) | Maximum burial depth (m) | Tpeak burial 3D model (°C) | Maximum burial depth 3D model (m) |
|---|---|---|---|---|---|
| BO1.0 | 0.52 | 83 | 1,970 | 93 | 1,550 |
| BO2.0 | 0.48 | 76 | 1,730 | 83 | 1,400 |
| BO3.0 | 0.70 | 107 | 2,770 | 116 | 2,100 |
| BO4.0 | 0.87 | 124 | 3,330 | 127 | 2,440 |
| BO5.0 | 1.51 | 169 | 4,830 | 163 | 3,300 |

Collectively, the initial findings from geochemical and logging data characterize the Amaltheenton-Fm as a regionally relatively homogeneous claystone formation within the study area. Thermal maturity quantifications through different methods corroborate variable maximum temperatures and consequently burial depths, following a SE-NW directed trend. Both findings are vital for the MATURITY project and highlight the suitability of the Amaltheenton-Fm for the stated objectives.



## 5.2 Systematic Changes in Petrophysical Properties of Amaltheenton-Formation

The thermal maturity range of the Amaltheenton-Fm measured at locations BO1 (0.50 VRr%), BO2 (0.48 VRr%), BO3 (0.70 VRr%), BO4 (0.85 VRr%), and BO5 (1.51 VRr%) is similar to that observed in nearby sections drilled by the old 1980's boreholes (Littke et al., 1988; Gaus et al., 2022). In their study, Gaus et al. (2022) also published an initial dataset on the petrophysical properties of the Amaltheenton-Fm, based on samples from the boreholes Wenzen, Dielmissen, Dohnsen, Harderode, and Haddessen. Porosities determined by He-pycnometry initially follow a decreasing trend with increasing thermal maturity, ranging from 14.69 % at 0.48 VRr% (Wenzen) to 7.67 % at 0.73 VRr% (Dohnsen), while bulk densities range between 2.32 g/cm³ (Wenzen) and 2.56 g/cm³ (Harderode). Despite strong uplift, the data presented by Gaus et al. (2022) is indicative to represent gradual compaction trends due to variable maximum burial depths. However, a slight increase in porosity is observed, reaching 8.27 % in the Haddessen borehole, with thermal maturity continuously rising to 1.45 VRr% while the bulk density is slightly decreasing to 2.48 g/cm³. Castro-Vera et al. (2024) attributed the trend towards increased porosities at the highest maturity to gas overpressures generated within the Posidonienschiefer-Fm, and resulting overpressure transfer to the surrounding formations.

A similar trend is observed in data from the MATURITY project boreholes from both, logging and lab data. Densities and porosities derived from the GGD logging signals for the boreholes BO1.1, BO3.0, and BO5.1 are presented together with respective lab data from the core material of all borehole location in Figure 14b,c,d. Regarding the thermal maturity, the Amaltheenton-Fm from those boreholes is similar to Wenzen (similar to BO1.0 and BO2.0), Dohnsen, Harderode, and Haddessen, respectively. Compressional wave velocities (Vp) usually increase with burial depth due to compaction and cementation processes. Our mean Vp values show a progressive increase in the order BO2.0<BO1.1<BO3.0<BO5.0<BO4.0. Vp values from those boreholes are plotted versus their respective thermal maturity quantified by vitrinite reflectance. A very good linear correlation (R²=0.99) between Vp and and vitrinite reflectance for normally pressured burial conditions (absence of overpressure) can be drawn from the boreholes BO1.1 to BO4.0 (Fig.14a), indicating a gradual progression of compaction and lithification due to tighter grain packing, cementation and consequently a reduction in porosity and permeability. However, the decrease in Vp between BO4.0 and BO5.0 is indicative for potential overpressure conditions in the latter. Vp values from location BO5.0 clearly fall out of the general, normal compaction trend. More evidence towards this end is also given by the development of log and laboratory derived density and porosity data. A decrease in bulk density accompanied by an increase in porosity and permeability can be observed in the Amaltheenton-Fm between boreholes BO4.0 and BO5.0 (BO5.1), despite the Amaltheenton-Fm accessed by borehole BO5.0 experienced approximately 900 m deeper burial depth during its burial history, following the 3D modelling results published by Castro-Vera et al. (2024). These trends in the investigated parameters contradict the general trend of increasing density and compressional wave velocity with greater burial depth while porosity decreases. A partly strong deviation from expected normal compaction trends is evident from all investigated petrophysical properties for a thermal maturity range somewhere between 0.87 VRr% and 1.51 VRr%, or related maximum burial temperatures and depths of 127 °C to 163 °C, and 2,440 m to 3,300 m.





**Figure 14: Vitrinite reflectance plotted against selected properties of the Amaltheenton-Fm from logging (rhombus) and lab (circle) data complemented by data from old boreholes after Gaus et al. (2022): (a) compressional wave velocity (Vp), (b) density, (c) porosity, (d) permeability.**



**5.3 Hydrogeological Borehole Conditions**

Gautschi (2017) compiled hydraulic conductivity (K) data derived from packer tests conducted at multiple locations within the Opalinuston Formation (OPA). These data, sourced from various studies, reveal a strong depth dependency of K (Fig. 15). In the uppermost decameters, particularly from shallow boreholes in the Swabian Alb (Germany) and the Lausen borehole (Switzerland), decompaction and weathering exert a clear influence, resulting in elevated K values as high as $10^{-4}$ m/s near the surface (Hekel, 1994; Gautschi, 2017; Vogt et al., 2017) (Hekel, 1994; Vogt et al., 2017; Gautschi, 2017). With increasing

depth, K values decrease rapidly by several orders of magnitude, reaching as low as $10^{-13}$ m/s below 28 m (Vogt et al., 2017). As overburden increases, the rate of this decline vanishes, as observed in boreholes Rinken and Benken (Gautschi, 2017; Vogt et al., 2017).

The hydraulic conductivities from the packer test series in boreholes BO1.0, BO2.0, BO3.1, and BO4.0 vary over two orders of magnitude. The lowest value, $1.15 \times 10^{-7}$ m/s, was recorded in BO4.0 at a depth between 24.8 and 31.9 m, which is consistent

with the K values for the Opalinuston Formation (OPA) compiled by Gautschi (2017). In contrast, the K values from BO1.0, BO2.0, and BO3.1 are notably lower than those reported for comparable depths (>63 m below ground level) in Gautschi's (2017) compilation. Hekel (1994) suggests that variations in local topography and geomorphological evolution may influence the thickness of the decompaction zone, potentially accounting for the higher hydraulic conductivities observed in BO1.0, BO2.0, and BO3.1.

Hydraulic conductivities measured from unconfined pycnometer tests on intact specimens taken from cores within the decompaction zone of boreholes BO1–BO5 range from $0.15 \times 10^{-12}$ m/s to $1.91 \times 10^{-12}$ m/s, seven orders of magnitude lower than the rock mass hydraulic conductivity at similar depth, and exhibit only a slight depth trend. Notably, the hydraulic conductivities of intact core specimens from the decompaction zone largely coincide with those obtained from packer tests and intact core specimens below the decompaction zone (Fig. 15).

This comparison supports the conclusion that hydraulic conductivity within the decompaction zone is primarily controlled by fluid flow through fractures. The observed decrease in rock mass hydraulic conductivity with depth, where the rock mass K approaches that of intact rock specimens, may be attributed to fracture closure induced by overburden pressure and possibly self-sealing processes. Notably, the hydraulic conductivity of intact rock remains largely unaffected by decompaction, falling within the range of values observed for both intact rock and rock mass at greater depths.





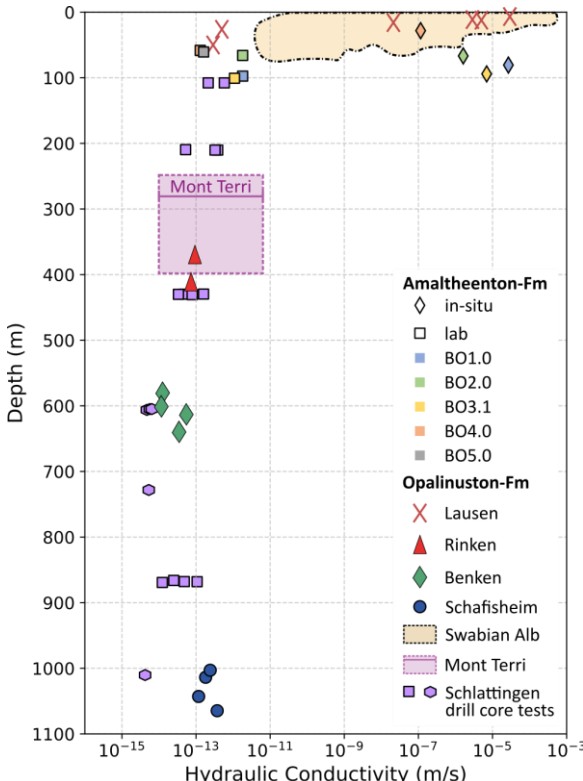

**Figure 15: Hydraulic conductivity vs. depth plot of the Opalinuston-Fm from boreholes in Northern Switzerland (Lausen, Rinken, Benken, Schafisheim, Schlattringen), the Mont Terri underground research lab, and the Swabian Alb (Germany) derived from packer tests and confined permeability measurements on core specimens (adapted from: Gautschi et al. (2017)) supplemented by lab and in-situ values of the Amaltheenton-Fm from borehole locations BO1-BO4. For data references of the individual boreholes see Gautschi et al. (2017).**

### 5.3 Outlook

The initial findings outlined above require further validation through more detailed investigations. To achieve this, comprehensive geochemical analyses, including X-ray fluorescence (XRF) and X-ray diffraction (XRD), will be conducted to assess the elemental and mineralogical composition of the Amaltheenton Formation at each location. These studies will quantify spatial variability, which is crucial for understanding basin-wide depositional patterns and potential lateral facies changes. Particular emphasis will be placed on rock properties critical for site selection. In this context, the geochemical-mineralogical study will primarily focus on determining the effective reactive surface area and its relationship to cation exchange capacity (CEC), both of which play a key role in radionuclide retention. Additionally, an extensive sedimentological analysis will be performed to identify specific facies zones, facilitating a more precise lithostratigraphic subdivision of the Amaltheenton Formation across different locations. Ongoing studies are also examining petrophysical and hydro-mechanical properties, including porosity, permeability, capillary pressure, strength, and elastic behavior. Further emphasis will be placed on characterizing the formation at the borehole scale. Logging data will be analyzed with a particular focus on fracture network





variability, enhancing the understanding of decompaction effects on hydraulic borehole conditions and their dependence on burial history. These detailed investigations will generate a unique and valuable dataset, significantly improving the understanding of burial-related changes in the physical properties of claystone.

## 6. Concluding Remarks

The MATURITY project aims to systematically investigate changes in physical rock properties related to variable burial history of the Upper Pliensbachian Amaltheenton-Fm in Lower Saxony (Germany). For this purpose, eight boreholes were drilled in the Hils- and Sack Syncline area which is part of the southwestern margin of the Lower Saxony Basin. In this contribution, we present initial results and insights into formation characteristics that were evaluated based on drill core material and in-situ investigations. In the future, detailed parameter studies will reveal the complex dependencies and interactions of burial-induced changes in mineralogical, petrophysical, mechanical, and hydrogeological claystone properties important for HLW disposal.

We demonstrate that the basic mineralogical composition of the Amaltheenton-Fm can be classified as argillaceous with a dominance in mixed-layer clay assemblage. This mineral composition shows only minor variability across the investigation area. Thermal maturity increases from the southeastern to the northwestern part of the study area. This trend could be confirmed based on different parameters, including vitrinite reflectance (VRr ranging between 0.48 VRr% and 1.51 VRr%) and is inferred to be induced by variable maximum depth reached during the burial history. The homogenous mineralogical composition and the gradually increasing trend of maximum burial are the two key prerequisites to conduct detailed parameter studies on the relationship between burial history and changes in the physical rock properties.

Initial results derived from FWS logs indicate a normal compaction trend for the Amaltheenton-Fm at locations BO1, BO2, BO3, and BO4, while Vp data from BO5 suggests the existence of overpressured conditions. This is underlined by density and porosity evolution across the area of investigation obtained from geophysical logging and laboratory data.

First hydraulic double-packer tests in four boreholes (BO1.0, BO2.0. BO3.1, and BO4.0) delivered permeability ranges of borehole intervals, spanning several orders of magnitude, suggesting hydrogeological conditions dominated by decompaction effects traceable to depths of 100 m. At the rock mass scale (meter to decameter), no significant correlations between the maximum burial depth and the hydraulic properties have been observed. However, significant changes in hydraulic transport properties with burial depth can generally be expected in the intact rock as evidenced on permeability data from all borehole locations. Future hydraulic borehole tests combined with log analysis regarding the fracture network will deliver more detailed information about hydro-dynamic behaviour and related transport characteristics on a meter to decameter scale.



Systematic changes of different physical rock properties such as compressional wave velocity, density, porosity, and permeability with increasing thermal maturity (Fig. 14) correlate with vitrinite reflectance. However, this trend is only valid for a maturity range between 0.48 VRr% and 0.87 VRr%, or related burial depths between 1,400 m and 2,440 m. A deviation from normal burial trends of those properties (see Fig. 1) was observed between thermal maturities of 0.87 VRr% and

1.51 VRr%. Under the assumption that the maturity sequence reflects variable maximum burial temperatures, significant changes in different physical claystone properties such as porosity, density and compressional wave velocity occur in an empirically derived temperature range between 124 °C and 169 °C, contradicting expectable, normal compaction trends (Fig. 16).

Upcoming studies will complement the previous analyses, to deliver an important data set for the further site selection process

in Germany. Eventually, that contributes to (a) provide a better comprehension on the complex processes that alter claystone properties during burial, and (b) facilitate the site and scale transferability of data by establishing transfer functions regarding the investigated parameters.

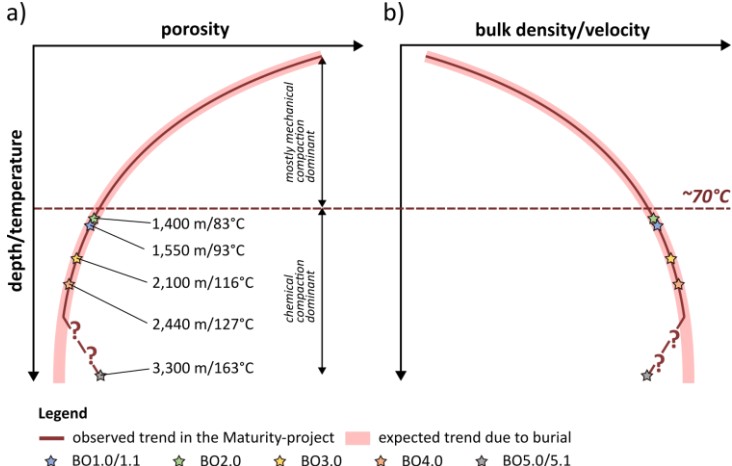

**Figure 16: Simplified and schematic depiction of expected and observed alterations in (a) porosity and (b) density and compressional**

**wave velocity induced by changing stress and temperature conditions during burial. Observed changes in properties follow the generally expected trends during burial for a thermal maturitiy range between 0.48 VRr% and 0.87 VRr% (BO1.0 to BO4.0) or related temperatures between 83 °C and 127 °C. A deviation for the same properties from the generally expected trend was observed for a thermal maturity of 1.51 VRr% (BO5.0) or a related maximum burial temperature of 169 °C. Temperature and property ranges not to scale. Burial depth and temperatures after Castro-Vera et al. (2024).**

*Data availability.* Data availability of presented data will be granted after publication.

*Author contribution.* RB, TS, GG, LW, MRJ, BMM, SG, RL, FA, LB, MCC: investigation. RB, RL: writing (original draft preparation). RB: visualization. LW, TS, GG, MRJ, RB, SG, FA, RL, JE, BMM, LB, MCC: writing (review & editing). FA, RL, JE: conceptualization, funding acquisition. FA, RL: supervision.

*Competing interests.* The authors declare that they have no conflict of interest.



***Acknowledgements.*** This project received funding from the Federal Company for Radioactive Waste dispoal and the Federal
Ministry for Economic Affairs and Climate Action, which is greatly acknowledged.

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
