# Peer review of "The influence of burial history on physical properties of claystones – Overview of a systematic research program across scales"

_EGUsphere, 2025_

## Author Comment (AC1)

**Referee #1**

Dear Dr. Minisini, thank you very much for your detailed revision work. We sincerely appreciate your constructive comments and thoughtful suggestions towards the improvement of our manuscript. We revised the manuscript following your valuable suggestions. All minor comments (i.e. typos, wording, and grammar) have been incorporated directly in the revised version. Comments that required deeper discussion are answered in detail below. All changes were made are with track changes on.

This paper focuses on rock units that could host disposal of radioactive waste. The paper prepares the ground for future contributions on the same topic and same area. Its main goal is to prove that the Amaltheenton Fm represents an optimal stratigraphic unit to study how a rock unit, originally having self-similar characteristics, is affected by different burial histories. In fact, the authors would like to investigate the impact of burial history on barrier-relevant properties of claystones ("low permeability, self-sealing efficiency with respect to fractures, sorption capacity, and mechanical properties for long-term stability of the underground infrastructure").

This is emphasized in passages like the following:

- "Collectively, the initial findings ... characterize the Amaltheenton Fm as a ... homogeneous claystone formation... Thermal maturity quantifications...corroborate variable ... burial depths, following a SE-NW trend. Both findings are vital for the MATURITY project and highlight the suitability of the Amaltheenton-Fm for the stated objectives."
- "The homogenous mineralogical composition and the gradually increasing trend of maximum burial are the two key prerequisites to conduct detailed parameter studies on the relationship between burial history and changes in the physical rock properties."

Although the authors explain how they understand the different thermal maturity in the area of study, they do not explain well how they understand that the rocks investigated in 5 wellbores are/were homogenous. The text says that the Amaltheenton-Fm is "a regionally relatively homogeneous claystone formation within the study area". Homogenous is a clear adjective that is not used with its significance in the paper. The quote itself says "relatively homogenous", which is an oxymoron. Actually it includes the term "regionally", as if zooming in, the formation would be heterogenous. In fact, the authors state that the Amaltheenton Fm "exhibits only slight variations between the individual borehole locations". But "slight" is not described in the text and their nebulous concept of "homogeneity" (among the 5 sites and within each site) rises a flag, as we are talking about the foundational work for potential nuclear waste disposal sites.

Sedimentary rocks are always variable, both in vertical and horizontal direction. However, as compared to lacustrine, fluvial, deltaic and coastal deposits, marine claystones can be regarded as homogeneous. Basically, the same Lower-Middle Jurassic sequence drilled (Amaltheenton, Posidonia Shale, Jurensismergel/Opalinuston extends from northern Germany towards southern Germany, Switzerland and France over large distances. Due to this fact, we call the units "homogeneous". However, there are variations, e.g. with respect to thickness of these units. A special aspect in the Amaltheenton are siderite concretions, often several cm thick, which occur randomly and thus cannot be predicted. Aside from this, there are only minor differences in clay mineralogy, organic carbon content, carbonate content etc. We expanded on these aspects in the paper and avoided the term "relatively" in combination with "homogeneous".

The authors could take advantage of the deep dives into these fine-grained rocks made by the Energy companies working on Unconventional resources. They might expand their views on fractures (permeability) and early diagenetic processes (porosity) affecting these claystones/mudstones.

Good idea, but based on German law, energy companies tend to keep their data confidential.

Since the authors themselves mention that they still need more "analyses....to assess the elemental and mineralogical composition of the Amaltheenton Formation at each location. ...and quantify spatial variability, which is crucial for understanding basin-wide depositional patterns and potential lateral facies changes", it seems this contribution might benefit from another round of readings by the authors.

There will be a special contribution, later, which will deal with mineralogy, elemental data, porosity etc. Keep in mind that this paper should provide an overview. However, we modified the sentence "need more analyses.."

Also, there authors should explain better to the reader the reason why the chose the Amaltheenton Fm (i..e, prove that the Amaltheenton Fm represents an optimal stratigraphic unit to study how a rock unit, originally having self-similar characteristics, is affected by different burial histories).

We agree and adapted the respective section as follows: "A claystone formation with a natural maturity sequence as result of differential burial and stable environmental conditions during deposition form the site requirements for the stated project objectives. Regional depositional consistency is important to ensure comparable mineralogical composition of the target formation, as mineralogy has influence on the parameters investigated along the course of the MATURITY project. Such conditions are given within parts of the Lower Saxony Basin (LSB), Germany. Various authors have identified strongly variable thermal maturity of Lower Cretaceous and Jurassic rocks linked to deep burial during the Late Cretaceous (Bruns et al., 2016). Notable maturity variations were observed for the Toarcian Posidonienschiefer-Fm on a relatively small regional scale in southeastern Lower Saxony, as documented (among others) by Koch & Arnemann (1975), Littke et al. (1988), and Mackenzie et al. (1988). However, the Posidonienschiefer-Fm itself is not regarded as a potential host rock for radioactive waste disposal due to its distinctive composition, characterized by a high carbonate and organic matter content, thus holding the potential for microbial activity, enhanced chemical reactivity, and oil and gas generation (Rullkötter et al., 1988). In contrast, both the underlying Upper Pliensbachian Amaltheenton-Fm, and the overlying upper Toarcian to Aalenian sequences, comprising the Jurensismergel-Fm and Opalinuston-Fm, are being evaluated as discrete sub-areas for potential host rock suitability (Fig. 3). Additionally, these units were uplifted to near surface levels by Cretaceous inversion tectonics, making them an ideal target horizon for shallow drillings within the scope of the MATURITY project. Apart from that the Jurensismergel-Fm/Opalinuston-Fm is located in the hanging wall, above of the Posidonienschiefer-Fm and has experienced strong and brittle deformations due to the degassing of the Posidonienschiefer in the eastern part of the transect, where thermal maturity reaches the gas window. Consequently, the Amaltheenton-Fm was selected as primary target formation suiting the project objectives."

Finally, although the authors focus on burial history, they mention other factors ("mineralogy, temperature, stress and water content") that affect the barrier properties they want to study. They should better explain why, among the several factors, they focus on burial history (beyond writing that "burial history and thus compaction affects porosity, permeability, and mechanical properties").

Our study cannot cover all aspects of nuclear waste storage in claystones. Here, the important aspect of burial and temperature history is mainly considered, because many Mesozoic and older claystones have been buried to great depth and later uplifted towards the surface. This burial history affects rock various properties important for barrier considerations. We revised parts of the introduction intending to introduce the reader to the choice of focussing on burial history related alterations. Also we added a subchapter introducing the MATURITY project objectives in a clearer manner.

Introduce why you focus on "claystones", in other words, explain why this is the most appropriate among the several different rock types (sandstones, limestones, evaporites, magmatic, metamorphic rocks).

As lithologies suitable for nuclear waste storage, claystones, salt (halite) and crystalline rocks (granite/gneiss) are considered. In our paper, we investigate claystones. Research questions regarding salt and granite/gneiss would be vastly different.

**52: why "up to 1 Ma"?**

Under the German Site Selection Act (Standortauswahlgesetz, StandAG), the proposed time frame for the safe disposal of high-level radioactive waste is at least 1 million years.

This is explicitly stated in the law as the period during which the containment of radioactive waste must be ensured without the need for human intervention, using passive safety provided by a deep geological repository.

We adapted the respective passage for clarity.

**53 and 92-94:** although most literature support the idea of "the deeper the less porous", the last decade dedicated to study the claystones/mudstones for Unconventional Resources demonstrated that some of these rocks get lithified before compaction, this understanding has implications for the idea of "the deeper the less porous". My colleague Macquaker and I wrote few weeks ago a piece about it in Marine and Petroleum geology and I hope this can help you with a different point of view on the same rocks we are analyzing in different industries. Maybe something to tie with your comments in lines 68-70. Also lines 75-79 may take in consideration the new literature derived from the Unconventional plays (it would be wonderful from their point of view if fractures would contribute to permeability, as they would produce more hydrocarbon resources; unfortunately the role of fractures does not go into that direction).

Thank you for the insightful comment. We agree that early cementation and lithification at shallow burial depths can significantly alter porosity-depth relationships. Fig.1a was designed to reflect this process (see arrow III), but we recognize that this was not sufficiently emphasized in the text. We have included a new sentence directly after the paragraph discussing causes of under-compaction (lines 68–70), where such deviations are addressed, and have cited Minisini et al. (2025) to incorporate this important perspective: "Another potential deviation from the general porosity—depth trend may result from early cementation during shallow burial, where chemical lithification occurs prior to significant mechanical compaction. This process has been increasingly recognized in studies of mudstones and claystones from unconventional resource settings, where early diagenetic cementation can "lock in" porosity and lead to complex, non-monotonic porosity—depth relationships (Aplin & Macquaker, 2011; Minisini et al., 2025)."

**71** -537: examples where nomenclature may be improved/refined. E.g. use "as muds are compacted to lower porosity mudstones..." 537: "as argillaceous mudstone". In sum, the rocks you are describing are mudstones (grains up to 64 micrometers, including grain size of clay -up to 4 micrometers- and silt -between 4 and 64 micrometers). They are not claystones (whose components are up to 4 micrometers). As an example, if a rock is composed by large clasts made 100% by clay minerals is not considered a claystone, it is a conglomerate.

We appreciate the suggestion regarding terminology and the distinction between mudstones and claystones. In our manuscript, the term "claystone" follows the lithostratigraphic nomenclature commonly applied in regional geological literature (e.g., Arp et al., 2021; Burnaz et al., 2024), where the Amaltheenton Formation is consistently referred to as a monotonous claystone unit. This terminology reflects not just grain size, but the mineralogical composition—with clay minerals (primarily illite and illite-smectite interlayers) consistently dominating the bulk rock (>55%, often >70%). Furthermore, high  $Al_2O_3$  concentrations and low MinC values support this classification. We therefore believe that the use of the term "claystone" remains appropriate for describing this formation in its geological context.

117: "mineralogically homogeneous claystone" explain the scale you are thinking about when using the term homogenous (e.g. bed scale, lamina scale, grain scale, cement scale, pore scale). Mention also the geological time encapsulated in the thickness of the formation taken in consideration (Amaltheenton Fm). This would make you wonder on the likelihood of "homogenous" environments of deposition (i.e. if the formation was deposited in 1 Million years, is it likely that the env of dep remained the same?)

We have revised the manuscript to clarify our use of the term "mineralogically homogeneous". In our study, homogeneity is considered at the formation scale, referring to both the spatial extent (approximately 50 km SE-NW directed transect) and vertical thickness (up to 90 m) of the Amaltheenton Formation, as sampled across five boreholes.

Specifically, we refer to bulk mineralogical composition, based on XRD analysis, which consistently shows a clay mineral content >55 wt.% at all locations and throughout the sampled depth intervals (see Figure 10a and Table 2; exceptions are siderite concretions). In comparison, the Opalinus Clay (the promising HLW host rock candidate in Switzerland) shows clay mineral contents between 43% and 73%. This deviation is linked to stratigraphic subunits (shaly versus sandy facies). However, the formation is also considered homogenous. Based on the first results of this study the Amaltheenton Fm has a bulk mineralogical composition similar to the Opalinus Clay. Also, the gamma ray logs show similar values across the individual boreholes and almost no fluctuations, indicating very consistent composition in horizontal and lateral extension.

Regarding the temporal aspect, we agree that the deposition period of the Amaltheenton Fm raises valid questions about environmental stability. However, detailed and recent paleoenvironmental studies by Burnaz et al. (2024) and Wijesinghe et al. (2025) indicate that the depositional environment during the Upper Pliensbachian was relatively stable, characterized by shallow marine conditions, high detrital influx, and oxic bottom waters. These findings support the assumption of a broadly consistent sedimentary regime, which is reflected in the mineralogical composition across our boreholes.

 Burnaz, L., Littke, R., Grohmann, S., Erbacher, J., Strauss, H., & Amann, F. (2024). Lower Jurassic (Pliensbachian—Toarcian) marine paleoenvironment in Western Europe: Sedimentology, geochemistry and organic petrology of the wells Mainzholzen and Wickensen, Hils Syncline, Lower Saxony Basin. International Journal of Earth Sciences. <a href="https://doi.org/10.1007/s00531-023-02381-8">https://doi.org/10.1007/s00531-023-02381-8</a>

- Wijesinghe, P., Littke, R., Burnaz, L., Blumenberg, M., Erbacher, J., Mann, T., Amann, F., & Bauersachs, T. (2025). Black shale deposition during the Early Jurassic: Geochemistry of Pliensbachian and Toarcian sedimentary rocks of the Hunzen Well, Hils Syncline, Northwest German Basin. The Depositional Record, dep2.70037. https://doi.org/10.1002/dep2.70037

118: "This formation should be i) accessible with shallow drillings (less than 100m)" explain why.

An explanation in this regard was added to the revised version. Drilling deeper than 100 m requires an application process which is much more complex according to German law. In addition, drilling deeper would be much more expensive.

119: "and ii) covered by rock strata to minimize the influence of weathering processes "Report the thickness of the "strata" covering the Amaltheenton in each borehole (use Table 1) and explain what is the assumed safety thickness to avoid weathering (if possible, reporting data supporting the thickness declared). Adjust with statement in line 165. Be ready to tackle the data from BO4.0 where the Amaltheenton is only covered by 6 m of Quaternary strata.

Thank you for this important suggestion. We have added a column in Table 1 indicating the depth to the top of the Amaltheenton Formation (mbgs) for each borehole. This provides a clear overview of the overlying strata thickness at each site. While there is no universally fixed threshold, several studies suggest that weathering in clay-rich formations typically affects the uppermost 0–30 m, depending on local conditions (see i.e. Studies from shallow boreholes in the Opalinus Clay: <a href="https://doi.org/10.1016/j.clay.2022.106793">https://doi.org/10.1016/j.clay.2022.106793</a>; <a href="https://doi.org/10.1007/s00767-017-0363-2">https://doi.org/10.1007/s00767-017-0363-2</a>, <a href="https://publikationen.uni-tuebingen.de/xmlui/handle/10900/56824">https://doi.org/10.1007/s00767-017-0363-2</a>, <a href="https://publikationen.uni-tuebingen.de/xmlui/handle/10900/56824">https://doi.org/10.1007/s00767-017-0363-2</a>, <a href="https://doi.org/10.1007/s00767-017-0363-2">https://doi.org/10.1007/s00767-017-0363-2</a>, <a href="https://doi.org/10.1

146: report the duration of the "safe long-term conditions" in years

See reply to comment no. 1.

**152:** northern not Northern

This was changed accordingly.

**Fig 3 and Fig 5:** Specify what formation the vitrinite maps refers to. If vitrinite refers to the Amaltheenton Fm, then note figure 1c of this article https://link.springer.com/article/10.1007/s00531-024-02477-9 showing a data point with 0.8 vitrinite in the Amaltheenton formation that does not match with your maturity map in fig 3. I see there is no discussion around that specific value, but hope you may find the information useful.

Thank you for this information. Please note that this specific data point is far outside of our own study area.

**232-248:** clarify better that all the data are referred to another formation, not the object of study. Add a sentence highlithing the imporantce of the data from the Posidonia for the overlying Amaltheenton formation.

We agree on your suggestion and have revised the section to explicitly state that these data are derived from the overlying Posidonienschiefer Formation and are not direct measurements from the Amaltheenton Formation. We added a sentence highlighting the importance of the Posidonienschiefer data for interpreting thermal maturity trends in the Amaltheenton due to their close stratigraphic and burial relationship. Note: the difference in depth by just a few tens of meters does not cause any significant difference in thermal maturity.

**250 and 591:** the measurements of porosity and permeability on rocks recovered more than 40 years ago ("core materials from the 1980's drilling campaign") rise a red flag. Rock was very likely dissecated and unsuitable for those measurements. Geologists and engineers working in Unconventional reservoirs have proven that those measurements are not reliable and so they spend millions of Euros/Dollars to collect fresh samples to have right values for their expectations on hydrocarbon production. Sealing radioactive waste might be considered more important than hydrocarbon production, hence this point should be clearly highlighted to the readers.

We fully agree that porosity and permeability measurements on core materials recovered more than 40 years ago must be viewed critically due to potential alterations caused by storage under ambient conditions. As such, we have added a sentence explicitly stating this concern and referring to the detailed discussion by Gaus et al. (2022), who carefully evaluated the extent of possible changes in the samples. Their study considered multiple lines of evidence, including preservation of pyrite, absence of gypsum, consistent bulk density values, and lack of significant macroscopic fracturing during drying, all of which suggest that the matrix properties of the core material remained largely intact. Nonetheless, we agree this limitation must be highlighted clearly and have done so accordingly in the revised manuscript. And keep in mind: our new cores were analysed in a fresh state to avoid any weathering after drilling.

253: "Their results align with the previous studies on the organic-rich Posidonienschiefer-Fm" If you investigate the range of values both for the studies made by Gaus et al. (2022) on the Amaltheenton, and for the studies made by Mann (1987) on the Posidonia, you'll discover that their different values represent different rock fabrics, in turn related to different environments of deposition. In this view, remember your previous statements on the "homogenous" environments of deposition (see comment on line 117).

This may be a misunderstanding. Of course, Posidinia Shale and Amaltheenton are completely different lithologies with different petrophysical properties. Therefore you can expect different values for the Posidonia Shale (Mann et al.) and Amaltheenton (Gaus et al.,), even if they are from the same borehole. We clarified this in the text.

**260-264:** are these statements conclusions from your work? Are they inferences? Or are they statements derived from previous studies?

We thank the reviewer for pointing this out. The statements in lines 260–264 are not conclusions or inferences from our own analyses but are derived from previous studies—specifically, Gaus et al. (2022) and Castro-Vera et al. (2024). To make this clearer, we have added the appropriate citation at the end of the paragraph. These referenced works provide the interpretations regarding overpressure

development and burial trends that help explain the observed deviations from typical porosity and density patterns.

**552:** are you sure that TOC "documents the increase in thermal maturity"?**

Yes, TOC changes with thermal maturity (see Rullkötter et al., 1988, Organic Geochemistry, for Posdonia Shale). During oil and gas generation, carbon is lost; roughly 50 % is transformed into liquid and gaseous products. However, this can be only tracked for rocks rich in organic matter (such as Posidonia Shale) and not for organic lean rocks such as Amaltheenton.

---

## Author Comment (AC2)

**Referee #2**

Dear Dr. Laurich, thank you very much for your detailed revision work and the constructive and helpful feedback, of which we believe that it will help to significantly improve our manuscript. We revised the manuscript following your valuable suggestions. Minor comments regarding style, grammar, and typos were deleted from this response letter for better overview. In almost all cases, we followed your suggestions. Changes were directly incorporated in the text. Below we answer your individual comments, which require deeper discussion in detail.

The authors report on current efforts within the research project MATURITY. MATURITY examines the relation of the thermal maturity of a clay formation to its physical rock properties. The authors state relevance of their work in eventually finding and comparing potential repository sites for radioactive waste.

The manuscript refers to samples of the Amaltheenton-Fm., gained from eight relatively close-by boreholes (~50 km) in the Lower Saxony Basin (Germany).

The authors state that the close-by sample selection ensures only marginal inter-sample divergence on mineral phases and consequently argue that the found variable physical rock properties must instead be caused by the variable maximum burial depth and uplift histories of the samples, with max. burial temperatures in the study region altering by up to ~80 °C. The authors claim a clear, continuous relation of VRr% to several physical rock properties. However, they also acknowledge a limitation of that claim for higher maturity cases (> 0.87 VRr%), where presumable location-specific hydrocarbon generation inversed the otherwise stated continuous porosity decrease with depth.

The burial history is referred to by current depth and by VRr% of the samples as well as by citing a regional 3D model of Castro-Vera et al. (2024), which seems the recent of several similar studies by the same Aachen-Group around Prof. Littke.

Properties that where related to depth derive either from borehole tests (among others: Th/K and Th/U ratios by gamma-ray spectr., vp by fullwave sonic, pbulk by gamma-gamma density and K and T by hydraulic packer tests) or from laboratory tests (among others: Tmax by Rock Eval analysis, mineral phases by Rietveld-XRD,  $\Phi$  by He-Pycn. and k by radial N2 uptake).

**General appreciation**

I suggest to accept this manuscript after a major revision.

The manuscript is generally well-written and the studied boreholes and samples make it a relevant and timely topic, surely interesting to many readers in the German site-selection procedure. It makes appetite for the next, currently gained results of MATURITY. Below, I outline several points that I would encourage the authors to revise.

**Risk of Misinterpretation**

The compelling question of a general reader could likely be: Which positive and negative effects come with increasing max. burial for the site-selection procedure? Can these help to discriminate one setting/region over another? I recognize that this manuscript, as correctly stated, focuses 'only' on the relation of max. burial to physical rock properties. Hence, to guide the general reader, I would hint that this relation is one of many essential criteria to consider in that compelling question.

Foremost, to me, that the effect of mineralogy and rock physical properties must necessarily be related to distinct boundary conditions. For instance, if a local shift to lower salinity pore water is allowed, that enhances the effect of swellable clay minerals, or, if uplift reduces the acting stress to be significantly lower than that of max. burial, it enhances brittleness (see studies on "over-consolidation ratio" (OCR) for this matter). Again, I acknowledge that this specification is beyond the scope of this study, I just fear that not stating its necessity can trick the audience into over-simplifications like: the deeper it has been (without early lithification or hydrocarbon generation), the less porous, the better for a repository site. In part, my fear is rooted by statements such as "[...] focus on [...] properties [...] referred to as barrier properties" (line 15ff, line 108ff and others, see line comments).

Thank you very much for these important considerations. We acknowledge that it is important to prevent potential over-interpretations of our findings and to clarify that the relation between maximum burial and investigated rock properties represents only one aspect relevant to host-rock assessment. In the revised manuscript, we have therefore (a) emphasized in the Introduction that burial-related effects must be considered alongside mineralogical, hydrochemical, and stress-related controls, and (b) added a short paragraph in the Discussion and Conclusions highlighting that the observed trends are context-dependent and not universally beneficial or detrimental for repository performance. These additions are intended to guide the general reader and avoid oversimplified interpretations.

Please note that currently, efforts are taken to investigate these crucial points in individual and detailed studies. The present study serves first and foremost as introduction to the MATURITY project, on which upcoming work is based.

**Improving Conciseness**

1. Broader categorizations of this kind ("barrier properties") are frequently invoked but often lead to less precise formulations, without improving the clarity (see in-line comments).

We agree that these categorizations are unspecific and prevent precise formulation in places. We have revised the respective sections for enhanced clarity in the manuscript.

2. Similarly, the introduction gives broader, unspecific objectives, e.g. line 125 "These correlations will provide a particular emphasis for the applicability to potential site areas", which are sometimes clearly biased, e.g. line 126 "The thermal maturity shall serve as key proxy for the burial history." or line 671 "In the future, [...] studies will reveal [...]".

A formulation similar as below would give the reader a more clear line of argumentation:

Our study addresses four aims:

- report on first measurement results of MATURITY and on how and where they were recorded,
- comparison of borehole and laboratory derived measurement results,
- comparison of max. burial values from VRr% and 3D modelling,
- discussion of the influence of max. burial on physical rock properties.

These aims could be followed by truly specifically describing their benefit to the site selection procedure for a radioactive waste repository. For instance: Does the max. burial trend allow to extrapolate from shallow boreholes to deeper regions of the same fm. if belonging to the same depositional centre? How is that helpful in the site-selection?

Thank you very much for these important considerations. We agree and have revised the introduction with these points in mind. We specifically pointed out the role of burial induced alterations in claystone characteristics with respect to site-transferability of geoscientific data in the site selection procedure.

Additionally, we added a subchapter "1.1 MATURITY project outline" that gives a clearer overview of the overall MATURITY project objectives/investigations, specifically stating the aims of the present study in a clear manner.

3. I encourage to rename and reorder some sections.

"Site Identification"

It would be advisable to delete this section and integrate the essential information into the Introduction, Previous Studies, or Geology sections (ensuring that it is not redundant with material already presented there).

In addition, the phrasing "was selected" (line 164 and similar arguments) leaves unclear from what pool of sites this particular one was chosen. My impression is that the site and its maximum burial difference as well as its mineralogy were already known before developing MATURITY, and that the aim was subsequently framed to match the site. To be clear: if true, this does not undermine the relevance of the subject—the topic remains important—but the wording should be revised to better reflect this relationship. Maybe just state in the introduction: "For these aims, the Amaltheenton Formation in the Hills and Sack Syncline area provides an excellent study opportunity. First, the nearby well sites (~ 50 km) exhibit largely similar mineralogy, while their Tmax values differ significantly by up to 80 °C. Second, the formation is of particular interest because it is also considered a potential host formation for nuclear waste disposal (among several other formations; see Fig. 3 for those that are clay-bearing)."

"Drillings, Sampling, Borehole Installations" & "Laboratory investigations"

Commonly, these are collectively given under "Methods". If the authors fancy, they could consider an overview table that lists all the derived value types in the first column, a Boolean field for "Lab/Borehole" as well as, in a third column, the method(s) applied for that value type. Maybe wise to list properties first and then derived index parameters. The table can be followed by "Below we describe the drilling/sampling procedure first and subsequently explain each method of table X separately." Apart from this extra table, the methods part can be shorted as it repeats reasoning that is given in the introduction already (see line comments).

"Site Characterization"

This is "Results".

"Implications and Outlook"

This is the "Discussion" section, which would benefit from a clearer line of argumentation if it were subdivided according to aims 1–4 outlined above. At present, it is sometimes unclear what merely serves as a qualification of the measurements and what, from the large number of measured values, actually contributes to the maximum burial trend concept.

"Concluding Remarks"

This is "Conclusion". Again, the line of argumentation and the clarity would win, if the conclusions regarding aims 1-4 were addressed sequentially.

Thank you very much for these valuable suggestions. We have adapted the renaming as proposed and also implemented the important information of the previous "Site Identification" chapter to the Introduction. We further expanded the Introduction to give a better overview of the MATURITY project and the executed and planned investigations. We believe that the made changes, following your suggestions, improved the general flow and enhanced clarity and readability of the manuscript.

**Arguments concerning the content**

1. The introduction refers to a low-min. difference being an a-priori reason for site identification, while in the discussion section it partly reads like an additional aim not stated initially. XRD and Th/K, Th/U ratios are discussed as 'minor variability' (line 536) or 'relative homogeneity' (line 578). Yet, these phrases lack definition regarding dimension and value range. Without such clarification, the conclusions risk being vague. For instance, two samples with identical carbonate content may differ strongly in strength depending on whether carbonate occurs as finely distributed cement or as fossil clasts (cf. Klinkenberg et al., 2009). Is this now a "minor variability"? If the authors better clarify that they address explicitly the variability of mineral phases, then, in turn, they should also (1) explain why they consider a variability of "58-75 wt.%" in clay minerals minor (line 536) and (2) hint that a mineralogical uniformity does not necessarily grants physical rock properties to be equal. First, this holds true as it is microstructure that defines physical rock properties (including porosity, porosity distribution, cementation, etc.), not mineralogy. Second, it would be necessary to adept to environmental boundary conditions – a certain mineralogy can have different implications (based on distinct depth, fluid flux and fluid type, etc.). In other words: the relation of mineralogy to physical rock properties can be non-linear, with an interdependence on the state of other controls. The authors state a "complex interplay" themselves, yet they do not fully acknowledge this important circumstance.

Basin wide comparable and stable environmental conditions during deposition are evidenced from several studies (i.e. Burnaz et al., 2024; Arp et al., 2021; Merten et al., 2024, Wijesinghe et al., 2025). This environmental stability resulted in similar mineralogical composition with clay minerals being the single dominant mineral group. Our data supports these points; i.e. the gamma ray logging signals are sensitive to mineralogy as the probe measures the natural radioactivity of elements Th, K, U. Consistent logging patterns can be observed across the individual sites and vertical logging profiles in the individual boreholes

In comparison, the Opalinus Clay shows clay mineral contents between 43% and 73%. This deviation is linked to stratigraphic subunits (shaly versus sandy facies). However, the formation is also considered homogenous. Based on the first results of this study the Amaltheenton Fm has a bulk mineralogical composition similar to the Opalinus Clay. Also, the gamma ray logs show similar values across the individual boreholes and almost no fluctuations, indicating very consistent composition in horizontal and lateral extension.

We see the need for clarification in the manuscript regarding these points.

Also while similar bulk mineralogical composition is an important basis for many of the taken investigations, the mineralogy is also object of investigation itself in the course of the overall endeavour

MATURITY. As stated in the introduction, the mineralogy of claystones might undergo important changes along gradual burial that are strongly related to the corresponding temperature changes, thermodynamic stability of certain mineral phases, and pore water chemistry (e.g. potassium availability). As stated, one of the most important mineralogical alterations in this sense is the conversion of smectite to illite. This conversion will (a) alter swelling behaviour and therefore self-sealing characteristics as illite is much less swellable than smectite, (b) enhance brittleness by reduced plasticity (illite is stiffer) (c) alter sorption characteristics as surface area and charge change. These changes are currently investigated in individual studies.

2. Sample alteration / desaturation: Have the fresh samples been directly weighted at site? What is their water loss when unpacked in the lab? Are there differences from storage time (older sample sets vs. recent sample sets)?

Alterations in water contents are considered minor, since core storage was maintained under constant ambient conditions and core preservation (air-tight packing) was regularly checked. Water contents were measured before and after laboratory testing, indicating the extent of dewatering processes after core extraction. The respective data will be presented in detailed studies in the future.

3. Burial history: The burial trend is recognized for a distinct depth-window (1,400 m - 2,440 m), below which hydrocarbon generation invokes a slight trend inversion. The aims are claimed to help the site-selection procedure, yet such regions with hydrocarbon generation will explicitly be excluded in that endeavor. Or have I gotten this wrong? This hypothesis of hydrocarbon generation is also not stated in the conclusion. After reading, I feel unsure if uplift has happened or not. The section "geology" clearly says "yes" (line 190 and Figure 5), but in later manuscript parts, in particular in the discussion, the effect of uplift is not examined anymore, simply the max burial and the "expected" trend that goes "along" within the identified window (line 33). For uplift: Would a clay rock have a "memory" of its max. burial depth, keeping its properties until getting somehow overprinted? What could make that "memory"? Internal cohesion/cementation? What could cause that overprinting? Long-term unloading? (It cannot be short-term unloading as that has happened to all samples during retrieval). Does it remain enigmatic? In this regard: have the upper most samples really seen "under compaction" (line 34) or more unloading due to uplift than the other samples? I encourage the authors to examine the interplay of burial and uplift, not just the Tmax / max. depth values (see comment on OCR above). Or, a bit drastic, is the authors' line of argumentation deliberately challenging the OCR theory? If so, that should be discussed and underpinned by arguments.

Thank you very much for this detailed and constructive comment. Indeed, regions affected by significant hydrocarbon generation are excluded from repository site selection. In our study area, petroleum generation and expulsion occurred only in the overlying Posidonienschiefer Formation at the northern locations BO3, BO4, and BO5, whereas the Amaltheenton Formation itself is organic-matterlean and has not experienced hydrocarbon generation. Moreover, the Amaltheenton Fm in the investigated area lies at shallow present-day depths, well above the potential repository depth defined by the German StandAG ( $\geq$  300 m below surface) and it therefore serves here solely as a natural analogue for studying burial-related rock-property evolution.

We agree that uplift has a strong influence on the present-day properties of the formation. The Amaltheenton in the study area is strongly overconsolidated due to significant post-burial uplift. The rock matrix retains a "memory" of past maximum burial primarily through irreversible compaction,

cementation, and the resulting cohesion, as reflected in laboratory-derived porosity, density, and permeability trends that correlate with maximum burial depth. At the same time, uplift and associated stress release affected the rock mass behaviour at larger scales. This is evident from the hydraulic data, which show a pronounced divergence between field-determined hydraulic conductivities and those derived from laboratory tests. We interpret this as matrix-controlled properties being superimposed by volume expansion and stress relaxation during uplift.

We also thank the reviewer for pointing out that our discussion did not explicitly address uplift in relation to OCR theory. We have revised the manuscript to clarify that we do not question the OCR concept.

Your question regarding under compaction and unloading might be a misunderstanding. Undercompaction is one potential reason for the divergence of BO5 from the observed burial trends. It is associated with very rapid and deep burial and pore water being "trapped" resulting in the build-up of excess pore pressure and altered effective stress. In contrast, unloading results from the reduction in vertical effective stress.

I am not an expert in the local geology nor in the borehole design and logging. Hence, I have made only a few comments regarding these chapters.

I would like to thank the authors—the effort required to compile this dataset must have been considerable. I am confident that the manuscript has the potential to develop into a strong publication and will be of great interest to many readers, particularly within the radioactive waste community. I do not wish to remain anonymous.

All the best,

Ben Laurich

**In-line comments**

Title:

The title "The influence of burial history on physical properties of claystones – Overview of a systematic research program across scales" illustrates the difficulty to address the multiple aims. Moreover, "systematic" in what sense? MATURITY seems a many-methods endeavour, apparently not all designed to examine the burial history. Plus "across scales" what scales are meant? Sample to regional? Is this wording actually meaning the third aim above (VRr% vs 3D model data)?

We acknowledge the reviewer's concern and propose the following revised title, which more clearly reflects the scope and terminology of the manuscript:

"The influence of burial history on physical properties of claystones at different scales – Overview on a research program on Lower Jurassic shales"

L 14ff "barrier properties" - What comes of defining that term? I'd suggest to avoid it and state specifically k and CEC. What is your argument for the other rock properties to fall in that category? Is swelling not also a bad thing if it behaves uncontrollably at stresses higher than Sig3? Is high mechanical strength not also favouring brittleness and fractures? Is lower porosity not also emphasising thermal conductivity? Otherwise: is not every single property relevant to the barrier functionality? I do not repeat this concern in following occurrences.

Indeed, we agree that this formulation might be misleading. We revised the manuscript and avoided this term. We specifically revised the respective sections in the abstract and introduction to account for this by stating i.e.:

"In Germany, clay-bearing formations are under investigation to potentially host a repository for high-level radioactive waste (alongside rock salt and crystalline rock). Their intrinsic properties such as low permeability, self-sealing efficiency with respect to fractures, and sorption capacity provide promising conditions for long-term waste containment. However, these properties are dependent on numerous factors such as mineralogical composition, temperature and stress conditions, and water content. Among these factors, the burial history and thus compaction affect mineralogy, porosity, permeability, and mechanical properties."

**and**

"Their suitability to act as natural barriers is mainly due to favorable properties, such as very low permeability (down to 10-21 m2), preventing significant focused fluid flow and related advective mass transport of radionuclides in aqueous solutions (OECD & NEA, 2022; Fisher et al., 2023). Additionally, their nuclide sorption capacity and self-sealing behavior mitigate the risk of radionuclide migration into the environment (Bastiaens et al., 2007; OECD & Nuclear Energy Agency, 2022). However, there are numerous factors influencing these key properties and the related sealing integrity of potential host rock formations, posing considerable complexity to site selection procedures. Among other factors such as mineralogical composition and pore water salinity (Bonin, 1998; Dewhurst et al., 1999; Carcione et al., 2019), the burial history exerts important controls on porosity, bulk density, permeability, and mechanical properties such as strength and elasticity (Jones & Addis, 1985; Bjørlykke & Høeg, 1997; Dewhurst et al., 1998; Czerewko & Cripps, 2006; Cripps & Czerewko, 2017; Ewy et al., 2020)."

**L 21 "multidisciplinary" would be more precise, if the authors name the aims 3 and 4 above. What is "across scales" referring to?**

What was meant here is the comparison between a small scale as investigated based on laboratory methods (cm) and the rock mass scale as investigated based on field methods (meters to decameters). We acknowledge, that this was not clear from the phrasing. The section was adapted accordingly.

**L 22 What thickness has the fm.?**

The formation was not entirely penetrated. However, borehole BO4.0 penetrated the Amaltheenton-Fm from few meters below ground level to its final depth of 94 m, i.e. about 90 metres.

**L 23 "Eight boreholes" for the claim of comparing max burial, it seems more important that 5 locations were examined.**

We agree that in this context the five locations are more important than the eight boreholes. The section was adapted accordingly.

**L 46 "Physical and .." – delete; circular reasoning (and vague "barrier attributes").**

We agree. The section was changed to: "Among other factors such as mineralogical composition and pore water salinity (Bonin, 1998; Dewhurst et al., 1999; Carcione et al., 2019), the burial history exerts important controls on porosity, bulk density, permeability, and mechanical properties such as strength and elasticity (Jones & Addis, 1985; Bjørlykke & Høeg, 1997; Dewhurst et al., 1998; Czerewko & Cripps, 2006; Cripps & Czerewko, 2017; Ewy et al., 2020)."

L 77 "control larger scale behaviour" – Counterargument: Fractures in clays have impermeable side walls. Calcite veins prove to have isotope signatures unrelated to close-by Ca-fossils that microstructurally often seem intact. Moreover, tracer profiles across faults (e.g. Main Fault in Mont Terri) show now deviation from an diffusion profile.

We generally agree, that fractures at a large scale do not necessarily need to enhance fluid flow, especially in low to moderately indurated formations such as OPA or COx, where self-sealing especially due to swelling of clay minerals is efficient. However, strongly indurated formations tend to lose the attribute of effective and rapid self-sealing due to swelling as result of clay mineral conversion with increasing temperatures during burial. This will also result in increased brittleness, facilitating fractures to form while the loss of swellable clay mineral phases prevents effective self-sealing. In such cases induced fractures might remain open and accessible for fluid flow, ultimately leading to pronounced scale effects between the low permeable matrix and the higher permeable rock mass (see i.e. Mazurek et al, 2009: Natural Tracer Profiles Across Argillaceous Formations: The CLAYTRAC Project). However, we adapted the wording to account for those differentiations. A dedicated study that investigates these kind of phenomena is currently on the way.

L 83 "self-sealing" this is an often missed opportunity to state what the term actually means. With ductility, as stated here, the fm. is "self-sealing" in the sense that it hinders larger brittle fractures to form by easily giving in to stress in a viscous manner. This is strictly different to "self-sealing" of fractures, where clay swelling CAN play a crucial role if the fm is not already saturated and/or if a pore fluid change to lower-ionic strength is at play.

Thank you very much for this important comment. Indeed, the ductile behaviour of soft clay and claystone such as Boom Clay is less prone to the formation of larger fractures compare to indurated claystone. This property should be clearly separated from self-sealing processes, i.e., due to clay mineral swelling. However, it is also shown (i.e., Bastiaens et al., 2007:DOI 10.1016/j.pce.2006.04.026) that effective self-sealing as a result of clay mineral swelling occurs faster in soft Boom Clay compared to indurated claystones such as Opalinus Clay. We agree that the section would benefit from further explanation and was adapted by stating: "Shallowly buried, soft, soil-like clays and claystones such as Boom Clay in Belgium hold advantages in terms of sealing integrity as the ductile behaviour of the material prevents larger brittle fractures to form easily. In addition, self-sealing as a result of clay mineral swelling allows for fracture closure within short time spans (Bastiaens et al., 2007). With progressive induration along deeper burial, claystones become more brittle promoting fracturing and potentially enhancing preferential fluid pathways, e.g., in an excavation damage zone (Neuzil, 1994; Bossart et al., 2002, 2004). Fracture self-sealing due to swelling might be effective but occurs over longer time-spans (Bastiaens et al., 2007; Bock et al., 2010)."

L 89ff See Rutter et al. 2001 for a good differentiation on what controls the mechanical behaviour, to become more precise here. Applies to L 93ff, too. (Evirn. controls vs. material intrinsic ones)

Thank you very much for this suggestion. We included the paper as important reference but do not see the need for further discussion as this study does not predominantly focusses on mechanical properties.

**Fig. 3 – inset refers to Fig 4, should be Fig 5**

Thank you very much of making us aware of this. The figure was adapted.

L 109ff "general changes" & "systematic and quantitative studies", "set the outlines for a series of detailed parameter studies designed to improve our understanding of [...] critical claystone properties" — all seem unnecessary broader categorizations, hindering to be specific and concise.

We agree. The section was changed to:

"Along the site-selection process, it is common practice to transfer data and information (e.g. parameters, investigation techniques, conceptual modes) across sites (Mazurek et al., 2008). The transfer of geoscientific data needs to account for the mutual and complex (inter-) dependencies between claystone properties and their burial history as they hold the potential of significantly influencing mechanical, hydraulic, and sorption characteristics even within the same formation. Hence, systematic studies on the depth- and temperature-dependent progression of these changes are needed that help to quantify the induced alterations and benefit data transfer across sites. However, to present day such studies remain scarce, as obtaining representative samples across different stages of diagenetic maturity typically requires costly and logistically demanding deep drilling campaigns. The MATURITY project, launched in 2022, seeks to fill this gap through a field-to-lab-scale research initiative carried out within the Amaltheenton-Fm, a Lower Jurassic (Late Pliensbachian) organic matter-lean marine claystone. In the following subsection the general project framework is outlined together with the objectives of the present study."

**Fig. 5 b) can the well be indicated, so to help spot the target formation? Maybe also indicate Amaltheenton fm as an indented annotation to "Lower Jurassic" in the legend.**

Yes, well locations were now indicated in the cross section. However, please note that the cross section was adapted from former studies (Jordan, 1989 and Wiese and Arp, 2013) and does not cut one of the current drilling locations. The changes made are therefore only indicative.

**L 245 So Rybacki is contradicting the authors hypothesis? What might he have over looked?**

According to Rybacki et al. (2014) UCS measurements document a peak for Harderode and smaller values in Haddessen and Wickensen. Since they used only three locations they stated no trend is apparent. They saw the same for Triax at 50 or 100 MPa Pc. Wic (low) - Har (high) - Had (low). This follows the expectable peak at Harderode. Accordingly, the findings by Rybacki et al. follow the expected trend. We corrected the section accordingly.

**L 248 So Mann & Müller are contradicting studies cited just above, where sec. porosity increases with maturity and hence must lower the density?**

The density from logging data (GGD and NN) reported by Mann & Müller increases with increasing thermal maturity across the borehole locations Wenzen, Diemissen, and Haddessen. The data from our study follows the same trend. However, we also derived data from the newly drilled boreholes BO4.0 that lies in close proximity to the old Harderode borehole (no logging data in Mann & Müller). Here, we observed the highest density while a gentle decline was observed towards BO5.0 which lies next to the Haddessen borehole.

L 256ff There is a study by Eseme et al. 2007, that examined the mechanical properties of Oil Shales. Eseme, E., Kroos, B. M., Littke, R., & Urai, J. L. (2007). Review of Mechanical Properties of Oil Shales: Implications for Exploitation and Basin Modelling. Oil Shale, 24(2), 159. https://doi.org/10.3176/oil.2007.2.06

This is a valuable study that reviews mechanical properties of oil shales. However, it does not fit the content of the respective section, as this section explores studies specifically executed in the area of investigation.

**L 347 list also Sr reduction**

Could you specify what Sr stands for? Do you mean saturation ratio?

**Table2: Can Pyrite be listed, too? That is crucial in weathering**

We agree, that Pyrite is crucial in weathering. A former study by Littke et al. (1993) was exactly about the strong effect of weathering on pyrite, while organic matter is less affected. However, at depth below 5-10 m pyrite is fresh in the Hils Syncline and not affected by weathering. See also the Burnaz et al. and Wisinghe et al. papers, where sulphur data are presented and discussed for two of our five locations. A dedicated publication is currently prepared that will target detailed mineralogical composition of the Amaltheenton-Fm.

**Fig. 13b: delete "gamma" from x-axis – correctly given in legend**

Done

**L 593 food for the discussion: Is this revesible? Is porosity decreasing after a P-depletion?**

Porosity always decreases with pp dissipation/increase in effective stress in a poro-elastic medium. The absolute value of porosity decrease depends on the bulk modulus and the magnitude of pp reduction. The increase in porosity is explained by overpressure generation (i.e. decrease of effective stress that may even lead to the formation of more porosity due to hydraulic fracturing); if dissipation is super slow during uplift, excess pp will remain and further reduce effective stresses.

**L 611 "A partly strong deviation" – reword.**

The sentence was adapted: "In fact, all investigated petrophysical properties for a thermal maturity range somewhere between 0.87 VRr% and 1.51 VRr%, or related maximum burial temperatures and depths of 127 °C to 163 °C, and 2,440 m to 3,300 m (from Castro-Vera et al., 2024) deviate from trends expectable along gradual compaction and cementation."

L 611 what is expected normal trend? This formulation occurs at multiple occasions. Is it and , as mentioned in the introduction? For OPA at Mont Terri, for instance such a deviation from "expected normal" due to cementation has also been reported by Corkum et al. (2007).

Corkum, A.G., Martin, C.D., 2007. The mechanical behaviour of weak mudstone (Opalinus Clay) at low stresses. International Journal of Rock Mechanics and Mining Sciences 44, 196–209. https://doi.org/10.1016/j.ijrmms.2006.06.004

The normal trend refers to a continuous change in rock properties, specifically visible in porosity and bulk density/volume with gradual burial as i.e. indicated in figure 1 and as identified by Athy (1930) as a result of compaction. We introduced this by referring to figure 1 on first occurrence.

L 633 Give a definition of "decompaction zone". Is this what is meant elsewhere by "undercompacted"? Again, decompaction / unloading alone can lead to fracturing if a certain brittleness is given.

We added a short section in the introduction chapter that introduces the decompaction zone early in the manuscript.

Undercompaction means a deviation from a normal compaction-depth trend as result of excess pore pressure working "against" the compaction during burial due to rapid burial and fluid retention.

L 635 see comment above on "memory", as all samples are tested unconfined and reveal the same values. Or do the authors imply that K is low with all sorts of samples always and hence laboratory K-measurements can be neglected in the site-selection procedure? Maybe illustrate: If a flux of X Bq is allowed to leave, how large, given the K values would a corresponding area need to be? How would that change in size for the different K values?

Thank you for this valuable comment and the opportunity to clarify our interpretation. We do not imply that laboratory-derived hydraulic conductivity values can be neglected for site-selection purposes. On the contrary, K values obtained from intact core samples are essential for quantifying the matrix-controlled transport capacity of claystones under repository-relevant conditions.

Our findings indicate that the laboratory-measured K follows the porosity trend with burial (BO2 > BO1 > BO3 > BO4 < BO5), suggesting that the rock matrix retains an imprint ("memory") of maximum burial and compaction. This "memory" refers to irreversible structural consolidation, not to variable K values among unconfined tests. In contrast, the hydraulic conductivity of the rock mass differs significantly where decompaction and stress-release fracturing occur, spanning several orders of magnitude above laboratory values. Therefore, while laboratory K adequately represents matrix behavior where the rock mass is unaffected by decompaction, it may underestimate transmissivity in shallow or strongly uplifted zones.

To address this point, we revised the manuscript to explicitly clarify this distinction.

L 643 "self-sealing processes" - (1) overburden pressure is a sealing process (sensu Dehandschutter et al. and Bock et al.), (2) if I got it right, then at this location fractures in deeper levels are not closed by overburden but rather have not developed in the first place, see "decompaction zone" above.

We agree that increase in overburden stress is considered a self-sealing process. To avoid misunderstandings the sentence was changed to: "The observed decrease in rock mass hydraulic conductivity with depth, where the rock mass hydraulic properties approaches those of intact rock specimens, may be attributed to fracture closure"

L 656: 'properties particular for site selection' — This phrasing suggests a hierarchy of importance. Which property is considered more important than the others, and by whom was this order established? Perhaps the authors mean to say that they 'address those parameters which, in their view, would most severely compromise the repository's suitability if they fall outside a certain range.' If so, that would still require argumentation.

We appreciate your comment and fully agree that our original phrasing could be interpreted as implying a predefined hierarchy of site-selection criteria, which was not our intention. No such hierarchy has been established in this study, and we do not intend to assign differing levels of importance to the investigated parameters. Our intent was to emphasize that, within the MATURITY project, certain rock properties are analyzed with a focus on their relevance to host-rock assessment and long-term safety functions, particularly those influencing hydraulic isolation and retardation capacity (e.g., cation exchange capacity, reactive surface area, and porosity). These parameters are

not "more important" in a general sense, but they are especially sensitive to diagenetic evolution and therefore provide valuable insights into how burial history may impact repository performance indicators.

To clarify this, we revised the text to explicitly state that our emphasis is on properties relevant to host-rock performance and safety assessment rather than implying any ranking.

**Conclusions:**

This chapter can again be more specific, sticking with well-defined aims and avoiding boarder categorizations. For example "[...]physical rock properties such as[...]", can be avoided by directly mention "the E-modulus, density, porosity and permeability values (or what others were correlated?) show a linear relationship to VRr% for the range of [...]".

We agree that greater specificity is beneficial and have revised the section accordingly to directly list and discuss the measured physical properties and their correlations. Furthermore, we reworked the conclusions with regard to the stated objectives in section 1.1.

L 689 All specimen have been tested for k unconfined. Discuss, if there will be a trend if subjected to depth-respective confining pressures.

This is an important part in a currently running study. We prefer to keep this aspect out of this paper.

L 689 and others: The authors seem a bit cautious in the formulation of their own results, pointing to the need for confirmation by ongoing MATURITY studies. However, they do not state what hypothesis or theory of this manuscript is going to be tested. If the upcoming works are rather (more detailed) repeated measurements, than there seems no need to be cautiously referring to them.

We appreciate your comments regarding the cautious formulation of our finding and agree that the respective sections can be rephrased. We want to clarify that we have strong confidence in the presented data and correlations derived from our current dataset. The ongoing MATURITY studies aim primarily to supplement this foundation by adding new parameters such as permeability tested under varying confining pressures and fluids, as well as cation exchange capacity (CEC), reactive surface area, mechanical strength, and elasticity in order to enrich the understanding of the Amaltheenton's behaviour, especially with respect to burial and uplift history. Our revised conclusion makes clearer, more assertive statements on the confirmed relationships, while noting that ongoing work will extend and deepen these insights rather than fundamentally questioning them.

Figure 16: What results can be drawn from (b) that are not given from (a)? Delete (b)? (a): is "chemical compaction" a defined term? It does not "compact" in the strict sense, is "chemical diagenesis" better suited? What justifies the 70 °C line? Is that from a reference? I guess that the y-axis caption should be named "max. burial depth / temperature" with 163 °C in the figure and 169 °C in the caption being a mistake. Or do the authors mean current depth / temperature? If so, then this figure would not provide a conclusion to the studied aim.

Thank you very much for the valuable suggestions. We agree, that the y-axis label should be max. burial depth/temperature and adapted it accordingly. The horizontal line at ~70 °C marks the approximate onset temperature of (a) the illite—smectite transformation and (b) the transition from mechanical to chemically dominated diagenetic processes, including cementation and pressure-solution. These temperature thresholds are well documented for clay-rich formations (see i.e. Bjørlykke, 1998; Peltonen et al., 2009). We adapted the caption to explain this threshold. We prefer to keep both panels (a) and

(b), as porosity, bulk density, and velocity represent distinct yet complementary parameters that respond reversely to burial-related mechanical and chemical processes. Indeed, the temperature in the caption should also be 163°C. This was adapted accordingly.